



# Effect of deep convection on the TTL composition over the Southwest Indian Ocean during austral summer.

Stephanie Evan[1], Jerome Brioude[1], Karen Rosenlof[2], Sean. M. Davis[2], Hölger Vömel[3], Damien Héron[1], Françoise Posny[1], Jean-Marc Metzger[4], Valentin Duflot[1,4], Guillaume Payen[4], Hélène Vérèmes[1], Philippe Keckhut[5], and Jean-Pierre Cammas[1,4]

[1]LACy, Laboratoire de l'Atmosphère et des Cyclones, UMR8105 (CNRS, Université de La Réunion, Météo-France), Saint-Denis de la Réunion, 97490, France

[2]Chemical Sciences Division, Earth System Research Laboratory, NOAA, Boulder, 80305, CO, USA

[3]National Center for Atmospheric Research, Boulder, 80301, CO, USA

[4]Observatoire des Sciences de l'Univers de La Réunion, UMS3365 (CNRS, Université de La Réunion, Météo-France), Saint-Denis de la Réunion, 97490, France

[5]LATMOS, Laboratoire ATmosphères, Milieux, Observations Spatiales-IPSL UMR8190 (UVSQ Université Paris-Saclay, Sorbonne Université, CNRS), Guyancourt, 78280, France

*Correspondence to*: Stephanie Evan (stephanie.evan@univ-reunion.fr)

**Abstract.** Balloon-borne measurements of CFH water vapor, ozone and temperature and water vapor lidar measurements from the Maïdo Observatory at Réunion Island in the Southwest Indian Ocean (SWIO) were used to study tropical cyclones' influence on TTL composition. The balloon launches were specifically planned using a Lagrangian model and METEOSAT 7 infrared images to sample the convective outflow from Tropical Storm (TS) Corentin on 25 January 2016 and Tropical Cyclone (TC) Enawo on 3 March 2017.

Comparing CFH profile to MLS monthly climatologies, water vapor anomalies were identified. Positive anomalies of water vapor and temperature, and negative anomalies of ozone between 12 and 15 km in altitude (247 to 121hPa) originated from convectively active regions of TS Corentin and TC Enawo, one day before the planned balloon launches, according to the Lagrangian trajectories.

Near the tropopause region, air masses on 25 January 2016 were anomalously dry around 100hPa and were traced back to TS Corentin active convective region where cirrus clouds and deep convective clouds may have dried the layer. An anomalously wet layer around 68 hPa was traced back to the South East IO where a monthly water vapor anomaly of 0.5ppbv was observed. In contrast, no water vapor anomaly was found near or above the tropopause region on 3 March 2017 over Maïdo as the tropopause region was not downwind of TC Enawo. This study compares and contrasts the impact of two tropical cyclones on the humidification of the TTL over the Southwest Indian Ocean.



## 1 Introduction

Deep convection plays an important role in delivering water and other chemical constituents to the Tropical Tropopause Layer (TTL, ~14-19km altitude, Fueglistaler et al., 2009) and lower stratosphere regions. Two important pathways for trace gas transport from the surface to the tropical stratosphere are i) deep convective injection directly into the stratosphere (Danielsen, 1982; Dessler and Sherwood, 2003), ii) convective detrainment into the TTL followed by a slow ascent into the stratosphere (Holton and Gettelman, 2001). Moist boundary layer air is transported to the upper troposphere by deep convection with the main outflow region at about 13 km (Folkins and Martin, 2005). However very deep convection may overshoot the 18 km level into the stratosphere, injecting water vapor and ice crystals directly (Corti et al., 2008; Khaykin et al., 2009; Avery et al., 2017). Studies based on Eulerian cloud resolving models have shown that those overshoots can moisten the lower stratosphere due to evaporation of ice crystals (Dauhut et al., 2015; Frey et al., 2015). However, convection can also cool the cold point tropopause (CPT) (Kuang and Bretherton, 2004), which can enhance dehydration via in-situ formation of cirrus clouds. In fact, the net impact of deep convection on TTL humidity (e.g. moistening versus dehydration) depends on the initial pre-convection TTL relative humidity with respect to ice (RHi) conditions and size of the ice crystals formed in the convective updrafts (Jensen et al., 2007; Ueyama et al., 2018). In sub-saturated TTL air, condensed ice is not removed quickly enough to produce net dehydration. Recent studies based on Lagrangian models (Schoeberl et al., 2014, Ueyama et al., 2015) that include convection and cirrus clouds microphysics show that convection impacts TTL cirrus clouds and water vapor near the tropical tropopause by 10-30% (~1 ppmv). Therefore, they concluded that convection is significant for the moisture budget of the TTL and must be included to fully model the dynamics and chemistry of the TTL and lower stratosphere.

As the exact role of convection in hydrating/dehydrating the stratosphere is still under debate, additional accurate TTL observations and modeling work are still needed to quantify the overall impact of convection on TTL composition and climate. At the moment, a realistic representation of deep convection and its effects remains a challenge for most global scale climate models and numerical weather prediction models (NWP).

Our understanding of how deep convection controls TTL humidity and composition, to a large extent, results from experiments in South America, the Western Pacific and South-East Asia. The role of the Indian Ocean in the global climate system is less understood than that of the Pacific Ocean, which has been more intensively observed and studied.

The tropical Indian Ocean (IO) has seen an unprecedented rise in heat content and is now home to 70% of the global ocean heat gain in the upper 700m during the past decade (Lee et al., 2015). Liu and Zipser (2015) showed using radar observations from the Global Precipitation Measurement (GPM) satellite that deep convection greater than 15km (Figure 1 of Liu and Zipser, 2015) can occur over the South IO with dozens of systems reaching above 17 km. These systems are likely tropical





cyclones over the south-west IO or thunderstorms that are often observed over Madagascar during austral summer (Roca et al., 2002; Bovalo et al.; 2012).

Tropical cyclones are unique among tropical and subtropical convective systems in that they persist for many days and hydrate a deep layer of the surrounding upper troposphere (Ray and Rosenlof, 2007). Ray and Rosenlof (2007) used measurements from AIRS to assess the impact of tropical cyclones in the Atlantic and Pacific basins on the amount of water

vapor in the tropical UT. They showed that tropical cyclones can hydrate a deep layer of the surrounding upper troposphere by ~30-50 ppmv or more within 500 km of the eye compared to the surrounding average water vapor mixing ratios. In addition, a modelling study by Allison et al. (2018) for TC Ingrid (2013) in the Gulf of Mexico indicated overshooting convection within the cyclone and associated strong vertical motions that transported large quantities of vapor and ice to the lower stratosphere.

Using 11-year TRMM precipitation satellite observations, Tao and Jiang (2012) identified overshooting tops in tropical cyclones (above 14 km) and showed that the South Indian Ocean is the second basin after the Northwest Pacific in terms of total number of overshooting tops (cf. Table 2 of Tao and Jiang, 2012). Even though convection occurs predominantly over land in the tropics, overshooting convection in tropical cyclones contributes ~15% of the total convection reaching the tropopause (Romps and Kuang, 2009).

The location of Réunion Island (21ºS, 55ºE) is thus ideal to study tropical cyclone effects on TTL composition. Réunion Island was formally designated as a Regional Specialized Meteorological Centre (RSMC) - Tropical Cyclones for the Southwestern Indian Ocean (SWIO, 0-40°S, 30-100°E) by the World Meteorological Organization (WMO) in 1993. The RSMC Réunion Island is responsible for the monitoring of all the tropical systems occurring over its area of responsibility. The SWIO is the third most active tropical cyclone basin with an average of 9.3 tropical storms with maximum sustained

winds ≥ 63 km/h forming each year (Neumann, 1993). In the SWIO basin, a storm system is called a tropical cyclone when wind speeds exceed 118 km/h.

We take advantage of the position of Réunion Island in the SWIO to study tropical cyclones' influence on TTL composition (water vapor/ozone) during austral summers 2016 and 2017. Austral summer (Nov-March) is the ideal time to sample convective outflow from tropical cyclones or mesoscale convective systems forming near Madagascar.

The present work is organized as follows. Section 2 has a description of the data used in this study. Section 3 presents the model used to infer the convective origin of the measurements. Section 4 presents the water vapor/ozone distributions over Réunion Island during the two storm events and thermodynamics of the troposphere and TTL. Section 5 discusses the convective influence on the measurements as inferred from an analysis of Lagrangian trajectories. The results are discussed



in Section 6. Section 7 contains a summary of our study.

## 2 Data

### 2.1 Balloon data

Balloon-borne measurements of water vapor and temperature in coordination with ground-based instrumentation (lidars) started in 2014 at the Maïdo Observatory (21.08°S, 55.38°E) within the framework of the Global Climate Observing System (GCOS) Reference Upper-Air Network (GRUAN) network (Bodeker et al., 2016). The balloon sonde payload consists of the Cryogenic Frospoint Hygrometer (CFH) and the Intermet iMet-1-RSB radiosonde for data transmission. The iMet-1-RSB radiosonde provides measurements of pressure, temperature, Relative Humidity (RH) and wind data (speed and direction from which zonal and meridional winds are derived). The CFH was developed to provide highly accurate water vapor measurements in the TTL and stratosphere where the water vapor mixing ratios are extremely low (~2 ppmv). CFH mixing ratio measurement uncertainty ranges from 5% in the tropical lower troposphere to less than 10% in the stratosphere (Vömel et al., 2007b); a recent study shows that the uncertainty in the stratosphere can be as low as 2-3% (Vömel et al., 2016). The CFH and iMet-1-RSB measurements have high vertical resolution (5-10m) and are binned in altitude intervals of 200 m to reduce measurement noise. Here we present CFH measurements (water vapor mixing ratio and Relative Humidity with respect to ice, $RH_{ice}$) from 2 soundings performed in austral summers 2016 and 2017, when deep convection was active near Réunion Island (tropical cyclones Corentin and Enawo, cf. Figure 1). During austral summer, balloon launch planning is optimized using a Lagrangian forecasting tool. This allows the identification of air masses with a convective origin that can be measured at the Observatory, thereby maximizing local resources by only measuring when convectively influenced air masses will be sampled.

In addition to CFH measurements at the Observatory, weekly Network for the Detection of Atmospheric Composition Change (NDACC)/Southern Hemisphere ADditional OZonesondes (SHADOZ) ozonesondes (Thompson et al., 2003; Witte et al., 2017) are launched from the airport (Gillot: 21.06°S, 55.48°E), located on the north side of the island (the flying distance between the Maïdo Observatory and the airport is ~20km). The ozonesonde is flown with a Meteomodem M10 radiosonde that provides meteorological variables such as temperature, pressure, relative humidity and winds. In this study, the NDACC/SHADOZ ozone and temperature measurements are reported in 200 m altitude bins.

### 2.2 Water vapor lidar data

A Raman water vapor lidar emitting at 355 nm is operating at the Maïdo Observatory since April 2013 (Baray et al., 2013;





Keckhut et al., 2015; Vérèmes et al., 2019). Laser pulses are generated by two Quanta Ray Nd:Yag lasers, the geometry for transmitter and receiver is coaxial and the backscattered signal is collected by a Newtonian telescope with a primary mirror of 1200 mm diameter. 387 nm ($N_2$) and 407 nm ($H_2O$) Raman shifted wavelengths are used to retrieve the water vapor mixing ratio. Depending on the scientific investigations, specific filter points and integration times can be chosen. The raw

vertical resolution is 15 m. Data are smoothed with a low-pass filter using a Blackman window. Based on the number of points used for this filter to vertically average the data, the vertical resolutions are 100-200 m in the lowest layers, 500 m in the mid-troposphere, 600 m in the upper troposphere and 700-750 m in the lower stratosphere. In order to convert the backscattered radiation profiles into water vapor mixing ratio profiles, the calibration coefficient is calculated from water vapor column ancillary data: GNSS (Global Navigation Satellite System) IWV (Integrated Water Vapor). The description of

the calibration method and the total uncertainty budget can be found in Vérèmes et al. (2019).

At the Maïdo Observatory, the lidar provides 4 to 8 water vapor profiles per month. The calibrated water vapor profiles of Lidar1200 database extends from November 2013 to December 2017. The time slot of routine operations is around 19:00 to 01:00 (+1) local time but there are intensive periods of observation during field campaigns that allowed longer measuring span. The Raman lidar water vapor observations were validated during the MORGANE intercomparison

exercise in May 2015 (Vérèmes et al., 2019). During the MORGANE campaign, CFH radiosonde and Raman lidar profiles showed mean differences smaller than 9 % up to 22 km asl.

Here we used the Raman lidar measurements for two nights when the CFH sondes were launched at the Observatory (25 January 2016 and 3 March 2017). The lidar water vapor profiles correspond to an integration time of 239 min and 184 min for the nights of 25 January 2016 and 3 March 2017 respectively. The lidar water vapor profiles are interpolated to the same

200-m vertical grid used for the CFH data and are shown up to 14.5 km. The mean lidar uncertainties for the troposphere below this level are 10.5% and 8.7% for 25 January 2016 and 3 March 2017 respectively.

**2.3 Satellite data**

The brightness temperatures of the infrared (IR) channel at 10.8 μm of the geostationary weather satellite METEOSAT-7 have been used to provide the regional characteristics of deep convection over the Indian Ocean. The satellite centered at

57.5°E provided images for the Indian Ocean from December 2005 to March 2017.

Aura Microwave Limb Sounder (MLS) v4.2 water vapor and ozone data were included in the study to compare with the in situ measurements and to evaluate the spatial extent of the convective air masses measured at the Observatory. In particular we have used water vapor from the Stratospheric Water and OzOne Satellite Homogenized (SWOOSH) data set (Davis et al., 2016). The SWOOSH dataset contains monthly mean stratospheric water vapor and ozone profiles from several satellite

instruments for the period 1984 to present. The data are available on a 3D (longitude/latitude/pressure) grid. The SWOOSH





input data for the period August 2004 to present day correspond to measurements from the Aura MLS satellite. The MLS water vapor data are available on a pressure grid with 12 levels per decade change in pressure between 1000 and 1 hPa (e.g. the vertical resolution is ranging from 1.3 to 3.6 km between 316 and 1 hPa). The estimated accuracy for MLS water vapor decreases from 20% at 216 hPa to 4% at 1 hPa and is ~ 10% in the TTL region (150-70 hPa).

Cloud-Aerosol Lidar with Orthogonal Polarization (CALIOP) onboard Cloud Aerosol Lidar and Infrared Pathfinder Satellite Observation (CALIPSO) makes backscatter measurements at 532 nm and 1064 nm since June 2006. We use the Total Attenuated Backscatter coefficients $\beta'_{532}$ available from the CALIPSO V4.10 level 1 lidar data products. Following Vaughan et al. (2004), the attenuated scattering ratio $SR_{532}$ (Equation 3 of Vaughan et al., 2004) profiles are computed as the ratio of $\beta'_{532}$ corrected for molecular attenuation and ozone absorption and the molecular backscatter coefficient $\beta_m$. $\beta_m$ is calculated
using the number density of molecules from the GEOS 5 global model of the NASA Global Modeling and Assimilation Office (GMAO), and the Rayleigh scattering cross section given in the CALIOP Algorithm Theoretical Basis Document (ATBD, cf. Equations 4.13a and 4.14).

## 3 Model

The origin of air masses measured at the Maïdo Observatory were assessed using the FLEXible PARTicle (FLEXPART)
Lagrangian Particle Dispersion Model (Stohl et al., 2005). FLEXPART is a transport model that can be run either in forward or backward mode in time. FLEXPART was driven by using ECMWF analysis (at 00, 12 UTC) and their hourly forecast fields from the operational European Centre for Medium Range Weather Forecasts - Integrated Forecast System (ECMWF-IFS). In March 2016, ECMWF introduced a new model cycle of the IFS into operations with a grid-spacing of 9 km roughly doubling the previous grid-spacing of 16 km used since January 2010. The ECMWF model has 137 vertical model levels
with a top at 0.01 hPa since June 2013. To compute the FLEXPART trajectories, the ECMWF meteorological fields were retrieved at 0.50° and 0.15° and on full model levels from the Meteorological Archival and Retrieval System (MARS) server at ECMWF. The 0.50° fields were used to drive the FLEXPART model over a large domain configured as a tropical channel, i.e., the domain is global in the zonal direction but bounded in the meridional direction (at latitudes ± 50°). Furthermore, higher-resolution domains can be nested into a mother domain in a FLEXPART simulation. Thus, to have a
better representation of convective transport associated with mesoscale convective systems or tropical cyclones with a horizontal dimension on the order of a couple of hundred kilometers over the SWIO, we included a nest domain covering the SWIO region (cf. Figure 2). If a particle resides in the high-resolution nest, the ECMWF meteorological data at 0.15° from this nest are interpolated linearly to the particle position. If not, the 0.50x0.50° ECMWF meteorological data from the mother domain are used to compute the trajectories. Retrieving high-resolution ECMWF fields from the MARS server for
FLEXPART consists in several steps which are:



- retrieve the meteorological model data output from ECMWF (horizontal winds, temperature, humidity, surface fields)

- compute total and convective precipitation rates, sensible and latent heat fluxes from the surface

- calculate the vertical velocity from the continuity equation

Therefore, the ECMWF high-resolution vertical velocity field already contains a convective mass flux component from the Tiedtke scheme used in ECMWF. The convective scheme used in the ECMWF-IFS, originally described in Tiedtke (1989), has evolved over time. Changes made include a modified entrainment formulation leading to an improved representation of tropical variability of convection (Bechtold et al. 2008) and a modified CAPE closure leading to a significantly improved diurnal cycle of convection (Bechtold et al. 2014). Particles are transported both by the resolved winds and parameterized

sub-grid motions, including a vertical deep convection scheme. FLEXPART uses the convective parameterization by Emanuel and Zivkovic-Rothman (1999) to simulate the vertical displacement of particles due to convection. The results from model runs with and without cumulus scheme in FLEXPART have been compared to assess whether convective mass fluxes could be resolved in the higher-resolution nest domain. The results of FLEXPART runs with and without cumulus scheme look fairly similar (not shown) and thus here we will present only the model results with cumulus scheme turned off.

To determine the transport history of air masses sampled by balloon launches, a so-called retroplume was calculated consisting of 10,000 back trajectory particles released from each 1 km layers of balloon launches used in this study, and advected backward in time. The initial positions of the 10,000 particles were distributed randomly within each 1-km vertical layer and a 0.10°x0.10° longitude-latitude bin centered on the balloon location. The dispersion of a retroplume backward in time indicates the likely source regions of the air masses sampled by the in situ instruments.

**4 Results**

**4.1 Tropical storm Corentin (January 2016) and tropical cyclone Enawo (March 2017).**

Tropical Storm (TS) Corentin started to form on 19 January 2016, east of 70°E. The METEOSAT 7 IR brightness temperatures on 19 January 2016 at 11 UTC indicate a vast clockwise circulation with some organization (not shown), indicative of tropical cyclone formation in the SH. The strengthening of the northerly monsoon flow favored the deepening

of the system in the subsequent days. Corentin became a moderate tropical storm (10-min maximum sustained wind speeds of 65 km/h) on 21 January 2016 at 00 UTC and at that time the TS center was located at 14.93°S, 75.63°E, ~2200 km to the northeast of the island. TS Corentin continued to intensify on January 22 while moving towards the south (see best track on



Figure 1). TS Corentin reached its peak intensity on January 23 at 00 UTC with 10-minute maximum sustained wind speeds of 110 km/h and the pressure at the center was 975 hPa. On 23 January 2016, convection was strong around 10°S in the

Mozambique Channel and near TS Corentin, especially in the northern part of the system. On January 24, Corentin had weakened into a moderate tropical storm.  On 25 January at 18 UTC (time of the balloon launch at the Maïdo Observatory), the storm was located at about 2500 km southeast of Réunion Island, near 26.03° south latitude and 79.19° east longitude (Figure 1).

The Madden Julian Oscillation (MJO) was active at the end of February and during the first week of March 2017 with a

signal centered over Africa and the Indian Ocean. A monsoon trough was well defined all over the basin along 9°S. On 28 February 2017 at 10 UTC, a zone of disturbed weather formed around 6.5°S, 70.2°E (not shown) with the building of clockwise rotating movement inside the cloud pattern. Favored by the MJO active phase and the arrival of an equatorial Rossby wave, Enawo initially formed as a tropical disturbance on March 2 with 10-minute maximum sustained wind speeds ~ 40 km/h. Enawo intensified to a moderate tropical storm at 06:00 UTC on March 3. At the time of the balloon launch at the

Observatory (~3 March, 18 UTC), Enawo was a tropical storm located near 13° south latitude and 56.42° east longitude, about 900 km north-northwest of Réunion Island (Figure 1). It strengthened into a severe tropical storm cyclone on 5 March at 00 UTC and became a category 1 tropical cyclone at 12 UTC. TC Enawo continued to intensify while moving toward Madagascar. It became a category 4 tropical cyclone on March 6 at 18:00 UTC, with 10-minute maximum sustained winds at 194 km/h. Enawo reached its peak intensity at 06:00 UTC on March 7, with ten-minute maximum sustained winds at 204

km/h and the central pressure at 932 hPa. TC Enawo reached Madagascar's northeastern coast on March 7 at around 9:30 UTC and was the third strongest tropical cyclone on record to strike the island. After March 8, TC Enawo gradually weakened to a tropical storm while moving southward over Madagascar.

The two balloon launches at the Observatory on 25 January 2016 and 3 March 2017 were specifically planned using FLEXPART Lagrangian trajectories and METEOSAT 7 infrared images. The goal was to sample the convective outflow

from TS Corentin and TC Enawo as well as convection north of Madagascar on 24 January 2016.

**4.2 Climatological and monthly mean water vapor distributions.**

Figures 3 and 4 show MLS water vapor volume mixing ratios at 215 hPa and 100 hPa averaged over December-January-February-March (DJFM) 2004-2017, January 2016 and March 2017. These values were computed by averaging the SWOOSH monthly mean water vapor concentrations gridded on a regular pressure/latitude (resolution of 5°)/longitude

(resolution of 20°) grid. Here we compare the water vapor mixing ratios at 215 hPa and 100 hPa for January 2016 (lower left panel on Figures 3&4) as well as March 2017 (lower right panel on Figures 3&4) with the climatological values for the



December-March period (upper panels on Figures 3&4).

On Figure 3, when comparing the water vapor mixing ratio at 215 hPa in January 2016 to the one observed in March 2017, one can see that the upper troposphere over the SWIO was much moister in January 2016 than in March 2017 with three
distinct regions of enhanced water vapor over Central Africa, the Indian Ocean and the Maritime Continent. The mean water vapor mixing ratio at 215 hPa over the SWIO in January 2016 is greater by ~23 ppmv compared to March 2017. Interannual variability modes such as the El-Niño–Southern Oscillation (ENSO) can affect the TTL temperature, and thus, water vapor distribution. The NOAA Climate Prediction Center Ocean Niño index (ONI), which is based on SST anomalies in the Niño 3.4 region, was equal to +2.5 K in January 2016 versus +0.1 K in March 2017
(http://origin.cpc.ncep.noaa.gov/products/analysis_monitoring/ensostuff/ONI_v5.php). January 2016 corresponded to strong El Niño conditions (one of the strongest El Niño event since 1950 according to the ONI index) while March 2017 was associated with neutral ENSO conditions. The water vapor mixing ratios at 215 hPa in for January 2016 are in agreement with MLS DJFM climatological values of water vapor at 215 hPa for El Niño conditions (upper left panel on Figure 3). Overall during El Niño conditions, water vapor mixing ratios at 215 hPa are enhanced over the SWIO west of 80°E. Ho et al.
(2006) have studied the variations of TC activity in the South Indian Ocean in relationship to ENSO effects. During El Niño periods TC genesis was shifted westward, enhancing the formation west of 75°E and reducing it east of 75°E. Therefore, on January 2016 the peak of water vapor west of 80°E at 215 hPa may be related to an increase in convection associated with strong El Niño conditions.

The Quasi-Biennial Oscillation (QBO) also affects TTL temperatures and humidity (e.g. Zhou et al., 2001; Yuan et al., 2014;
Davis et al., 2013). Thus, we computed the climatological water vapor concentrations at 100 hPa over the SWIO according to the phase of the QBO. Following Davis et al. (2013), we defined a QBO index as the zonal mean (10°S-10°N) of the difference in the ERA-Interim zonal wind at 70 and 100 hPa. A positive QBO index ($u_{70hPa}$ - $u_{100hPa}$ > 0) corresponds to westerly shear conditions and the warm phase of the QBO (Baldwin et al., 2001). A negative QBO index corresponds to easterly shear conditions and the cold phase of the QBO (CPT temperatures are cooler during the easterly shear phase of the
QBO). The westerly shear phase of the QBO (QBO index > 0) is associated with slightly higher water vapor concentrations at 100 hPa over the SWIO during the December-March period. The MLS climatological values of water vapor mixing ratio at 100 hPa over the SWIO are higher for austral summers with westerly QBO shear by 0.43 ppmv (Figure 4). The mean January 2016 water vapor mixing ratio at 100 hPa over the SWIO is 4.2 pmmv versus 3.7 ppmv in March 2017 as compared to the climatological values of 3.51 ppmv for January and 3.44 ppmv of March. The difference of 0.50 ppmv between the
two periods cannot be explained by the phase of the QBO as both months corresponded to QBO westerly shear conditions (2.33 m/s for January 2016 and 4.79 m/s for March 2017). However, the higher water vapor mixing ratio at 100 hPa in January 2016 could be related to strong El Niño conditions as Avery et al. (2017) have reported large lower stratospheric (82





hPa) water vapor anomalies (~ +0.9 ppmv) associated with the strong 2015-2016 El Niño. In December 2015, SWOOSH water vapor mixing ratio anomalies of ~ +1 ppmv are observed over the Indian Ocean (not shown). In January 2016, the

anomalies over the SWIO have eased to 0.7 ppmv  (not shown).

**4.3 Water vapor/ozone profiles**

Figure 5 shows two CFH water vapour mixing-ratio profiles (black lines) taken at the Maïdo Observatory on 25 January 2016 at 17:50 UTC and 3 March 2017 at 18:00 UTC. The lidar water vapor profiles for those two nights are also displayed in green. The red and magenta lines correspond to NDACC/SHADOZ ozonesonde balloon profiles launched from Gillot on 18

January 2016 (purple line), 4 February 2016 (red line) and 3 March 2017 (purple line on the right panel). The ozonesonde data correspond to daytime measurements (balloon launches at ~11 UTC) while the CFH water vapor data correspond to nighttime measurements in order to coincide with water vapor lidar measurements at the Maïdo Observatory. Overall good agreement is seen between the lidar and CFH water vapor profiles over the whole troposphere. Note that the CFH water vapor profiles were not used to calibrate the lidar water vapor profiles as explained in section 2.2.

The altitude range 2-12 km on 25 January 2016 is moister by ~50% than the same altitude range on 3 March 2017 (mean water vapor mixing ratio of 5076 ppmv and 4375 ppmv between 2 and 12 km on 25 January 2016 for the CFH and lidar respectively versus 3335 ppmv and 3398 ppmv on 3 March 2017 for the CFH and lidar respectively). The austral summer season, with warmer temperatures and greater cloudiness, reaches its peak in January/February and this could explain in part the higher humidity observed in January than March. In addition, January 2016 corresponded to a strong El Niño period and

this could lead to higher tropospheric moistening associated with ENSO (Tian et al., 2019). On 3 March 2017, a moist layer was observed between ~12 and 16 km in both CFH and lidar water vapor profiles with corresponding low ozone values.  On 25 January 2016, a similar layer of moist air/low ozone is observed between ~ 9 and 14 km. The lidar smooths out the peak of water vapor at 10 km observed on 25 January 2016 but this could be due to the longer integration time used for that night (239 min). The CFH water vapor mixing ratio profiles have a minimum of 2.5 ppmv at 17.10 km (94 hPa) and 2.70 ppmv at

18.10 km (77.1 hPa) on 25 January 2016 and 3 March 2017 respectively.

Also shown in Figure 5 is the climatological mean ozone profile for DJFM 1998-2017 (blue lines on Figure 5a&b). Anomalously low mixing ratios approaching surface values are seen in the upper troposphere for both the 4 February 2016 (red line, Fig. 5a) and 3 March 2017 (purple line, Fig. 5b) ozone sonde flights. In the upper troposphere, the climatological mean ozone mixing ratios ranges from about 60 ppbv at 10 km to 100 ppbv at 15 km. There is a steep gradient above 17 km,

indicating the transition from troposphere to stratosphere. On 3 March 2017, ozone mixing ratios between 10 and 15 km are ~ 45 ppbv below the climatological values (mean value of 25.10 ppbv for the 10-15 km layer on 3 March 2017 versus 70.1



ppbv for the climatological ozone profile).

Between 18 January and 4 February 2016, ozone mixing ratios in the upper troposphere decreased by ~30 ppbv and are 38 ppbv below the climatological values on 4 February 2016. Tropical storm Corentin reached its peak intensity on 23 January 2016 at 00 UTC and its center was located 1735 km east of Réunion Island. These low ozone mixing ratios in the upper troposphere on 4 February 2016 were observed after the storm had its major influence on UT ozone, transporting air with surface ozone values upward via strong convection and mixing out into the larger environment. In comparison, the 18 January 2016 ozone profile was not influenced by TS Corentin. The lower ozone values on 3 March 2017 compared to those observed on 4 February 2016 could be explained by the fact that TC Enawo was closer to the island (~902 km north of the island), was still intensifying and was a stronger system than TS Corentin. Above ~17 km the ozone profiles on January/February 2016 and March 2017 are more similar to the climatological mean ozone profile, suggesting that deep convection did influence the upper troposphere but not the lower stratosphere. We will later show using FLEXPART that the moist/low ozone layers in Figure 5 are associated with the convective outflow of a mesoscale convective system north Madagascar on 23 January 2016, TS Corentin and TC Enawo.

## 4.4 Relative humidity and temperature profiles

Figure 6 shows the CFH profiles of $RH_{ice}$ (computed using the Goff-Gratch equation [Goff and Gratch, 1946] for water vapor pressure) on 25 January 2016 and 3 March 2017 as well as collocated CALIOP nighttime backscatter measurements. The CALIOP measurements shown on Figure 6 include only those within ±5° latitude and ±10° longitude of the Maïdo Observatory. The CALIOP measurements on 25 January 2016 correspond to a CALIPSO overpass east of the island around 4 hours after the balloon launch and the mean longitude difference between the CALIPSO overpass and the Maïdo Observatory is 2.4° for Figure 6-top. On 3 March 2017, the CALIPSO overpass was west of the island and also 4 hours after the balloon launch. The mean longitude difference between the CALIPSO overpass and the Maïdo Observatory is 5.3°. The latitude-height cross-section of CALIOP $SR_{532}$ on Figure 6 correspond to measurements with a 60 m vertical resolution. The horizontal interval of the CALIOP data along its orbit is 330 m; for this study we use a 9-point running average to reduce noise.

Figure 6 (top) shows significant structure in the $RH_{ice}$ profile measured on 25 January 2016. Higher values of $RH_{ice}$ (> 40%) between 13 and 15 km coincide with higher values of CALIOP $SR_{532}$ between 12 and 15 km. The $RH_{ice}$ reaches its maximum value at the coldpoint altitude (17.3 km). The CALIOP $SR_{532}$ indicates a cirrus cloud between ~12 and 15 km north of the island. The cirrus layer extends from ~16.2°S to 20°S corresponding to a horizontal scale of ~400 km. METEOSAT 7 infrared brightness temperature at 21:30 UTC, so ~10 minutes before the CALIPSO overpass at 21:39 UTC


on Figure 6 (top), indicates a large area of deep convection near 15°S and extending from ~ 50° to 75°E (not shown). The monsoon trough was located between 17°S/50°E and 14°S/70°E on 25 January 2016 which promoted deep convection and convective activity was also observed in the South-Eastern quadrant of TS Corentin. The cirrus cloud observed below 15 km on Figure 6 (top) was most likely from convective detrainment north of Réunion Island. The $RH_{ice}$ profile on January 25 indicates intertwined layers of dry air ($RH_{ice}$ less than 40%) at 7, 9, 12 and 16 km and less dry air ($RH_{ice}$ ~ 50%) at 8, 11, 15 and 17km. While convection north of Réunion Island around 15°S and TS Corentin had mixed the troposphere over the Southwest Indian Ocean, no cirrus clouds were directly observed on 25 January 2016 above the Maïdo Observatory. The layers of $RH_{ice}$ ~ 50% at 15 and 17 km may be due to convective detrainment. The cirrus cloud below 15 km detected by CALIPSO north of the island on January 25 indicates that deep convection detrained ice and water vapor in the upper troposphere north of the island. There was a northerly wind between 10 and 17 km on 25 January 2016 with a peak around -25 m s$^{-1}$ at 15 km (not shown). Moist air detrained by deep convection north of Réunion near 15°S may have been transported to Réunion Island in ~ 6 hours and during that time the moist air mass could have mixed with drier air, thereby explaining the layers of $RH_{ice}$ ~ 50% at 15 and 17 km on Figure 6. The origin of these layers has also been determined using the FLEXPART Lagrangian model, and the results are presented in section 5.

On 3 March 2017, a layer close to saturation ($RH_{ice}$ > 80%) can be observed between 12 and 16 km (Figure 6, bottom left) with $RH_{ice}$ up to ~ 100% at 12.5 and 14 km, below the coldpoint altitude (16.1 km). The altitude range 12-15.5 corresponds to cloudy air and a cirrus cloud can be seen in the CALIOP measurements of $SR_{532}$ between ~13 and 15 km extending from 18.4°S to 21.2°S (Fig 6, bottom right). Above Réunion Island, the cirrus is ~ 1.5km thick and the maximum thickness of ~ 3 km is observed north of the island at 20.5°S. A second cirrus cloud can also be observed below 15 km north of 17.4°S.

The CPT height is 16.10 km on 3 March 2017 while it is 1.2 km higher on 25 January 2016. The CPT temperature was 192.64 K on 25 January 2016 and 194.58 K on 3 March 2017. On 3 March 2017, the layer between 16 and 18 km is almost isothermal with a mean temperature of 195 K while the tropopause is sharper on 25 January 2016. A seasonal mean (December-March) temperature profile is computed for the period 1997-2017 using the NDACC/SHADOZ dataset. The weekly NDACC/SHADOZ launch is performed at the airport in the north part of the island (Gillot, 20 m a.s.l.). The flying distance between the Maïdo Observatory and the airport is ~20 km so while boundary layer temperature values will differ for the two sites, free troposphere/TTL temperature distributions can be compared as they are less influenced by topography. The seasonal mean CPT height is 17.31 km for the period December-March with a mean CPT temperature of 193.90 K (Table 1). The tropical tropopause is higher and colder during austral summer as a response to large-scale upwelling in the tropical stratosphere (Yulaeva et al., 1994) and convection (Highwood and Hoskins 1998). The iMet radiosonde temperature profiles are then compared to the seasonal mean NDACC/SHADOZ temperature profile. The upper panels on Figure 7 show temperature profiles from NDACC/SHADOZ and the iMet radiosonde. The black line shows the NDACC/SHADOZ



seasonal mean temperature profile while the red line corresponds to the iMet temperature profile observed at the Maïdo Observatory.

A large positive temperature anomaly is observed on 25 January 2016 over a broad tropospheric region from 2 to 16 km
(mean amplitude of +2.5 K) with a peak warming of +4.6 K at 10km. On 3 March 2017, a warm temperature anomaly is mostly observed between 6 and 14 km (mean amplitude of +1.1 K) with a peak value of +3.1 K near 12 km. The stronger warming of the troposphere observed in January 2016 may be due to the strong 2015/2016 El Niño. The connection between interannual variations in tropical tropospheric temperature and ENSO is well established (e.g., Yulaeva and Wallace 1994; Soden 2000). Using 13-year of temperature data from the tropospheric channel of the microwave sounding unit (MSU-2),
Yulaeva and Wallace (1994) showed that a tropospheric warming occurs almost uniformly over the tropics and that the magnitude of the warming is around 0.5-1°C for strong El Niño years. Chiang and Sobel (2002) updated the analysis of Yulaeva and Wallace to include the response to the strong 1997/98 El Niño (ONI of +2.2 K in DJF 1998) and indicated MSU-2 temperature anomaly of ~1.2 K in January 1998 (cf. Figure 1 of Chiang and Sobel, 2002). Note that the MSU-2 temperature data used in these studies provide a measure of the mean temperature of the 1000-200 mb layer (corresponding
to the surface to ~ 11 km using a scale height of 7 km). Thus, part of the strong tropospheric warming (especially in the lower part of the troposphere) observed in January 2016 may be due to the strong 2015/2016 El Niño (ONI of +2.5 K in DJF 2016). Assuming a tropospheric warming of ~ 1K in response to a strong El Niño, the magnitude of the upper tropospheric warming observed on 25 January 2016 (mean amplitude of 3.4 K between 10 and 14 km) becomes more similar to the one observed on 3 March 2017 (mean amplitude of 1.9 K between 10 and 14 km) if the effect of the 2015/2016 El Niño is
removed.

Figure 7 indicates cold temperature anomalies within 16-19 km above the tropospheric warm anomalies on 25 January 2016. The mean amplitude of the 16-19 km temperature anomaly is -1.6 K with a maximum cooling of -3.6 K at 18 km. A similar feature is observed on 3 March 2017, with a cooling between 14 and 17 km with a mean amplitude of -2 K and maximum cooling of -4.5 K at 15.1 km. The upper tropospheric warming and near tropopause cooling observed on both dates is
consistent with a temperature response to deep convection (e.g. Sherwood et al., 2003; Holloway and Neelin, 2007; Paulik and Birner, 2012). The cooling around the tropopause can be explained by either radiative cooling by cirrus clouds over the regions of deep convection (Hartmann et al., 2001) or diabatic cooling through convective detrainment (Sherwood et al., 2003; Kuang and Bretherton, 2004). CPT properties can also be modified by convectively driven waves (Zhou and Holton, 2002; Randel et al., 2003).

To assess the effects of deep convection on temperature in the upper troposphere and near the tropopause, we looked at the distribution of deep convective clouds in the days preceding the soundings. The location of deep convective clouds can be



assessed by using maps of METEOSAT 7 infrared brightness temperature. Figure 8 shows convective cloud coverage for the 3-day period preceding the sonde launch date at the Maïdo Observatory. Convective cloud coverage was estimated using 3-hourly METEOSAT 7 infrared brightness temperatures at 5 km resolution. A threshold of 230 K is used to detect deep
convective clouds in the METEOSAT 7 brightness temperature data (i.e. pixels with brightness temperatures less than 230 K correspond to convective clouds). This threshold has been previously used to identify convection on geostationary satellite infrared images (e.g. Tissier et al., 2016). This temperature corresponds to a height of about 11 km in the NDACC/SHADOZ climatological-mean summertime profile of temperature on Figure 7. Prior to 25 January 2016, the main deep convective activity is located ~1500 km north of the island between 50 and 70°E and around tropical storm Corentin. From 28 February
to 3 March 2017, convective clouds are located ~500 km north of the island and correspond to the intensifying tropical storm Enawo. The coldest cloud tops (≤ 190 K) that correspond to the deepest convection are indicated by red dots on Figure 8.

Paulik and Birner (2012) investigated the deep convective temperature signal based on SHADOZ ozone and temperature data. Low ozone concentrations in the upper troposphere are indicative of convective transport from the boundary layer. They looked at temperature anomalies corresponding to low ozone anomalies between 12 and 18 km, thus temperature
anomalies influenced by deep convection. A strong warming was observed near the level of main convective outflow at ~12 km and cooling was more pronounced above ~ 15 km and near the CPT at ~17 km. Thus, the upper tropospheric warm temperature anomalies as well as cold temperature above 15 km and near the tropopause on Figure 7 are coherent with a deep convective temperature signal. Paulik and Birner's study also showed that the amplitude of the temperature anomalies increases as convection strengthens with a warming of ~2K in the upper troposphere and a cooling of around -3K near 16 km
(cf. Figure 5 of Paulik and Birner, 2012). Using CloudSat observations of deep convective clouds and COSMIC GPS temperature profiles, they showed that the deep convective temperature signal (i.e. anomalously warm upper troposphere and an anomalously cold upper TTL) was only present for deep convective clouds above 15 km. Although the magnitude of the temperature anomalies decreases with increasing distance from convection, they observed a deep convective temperature signal during DJF ~3500 km away from the convective event. Within 1000 km of the deepest convection (deep convective
clouds above 17 km), the convective temperature anomaly exceeds 0.75 K in the upper troposphere and ranges from -1 K to -2.0 K near 16 km. In our case, the deepest convective clouds with cloud tops colder than 190 K are 1000 km away from the island on 22-25 January 2016 and are closer from the island at ~500 km on 28 February-3 March 2017. Although deep convective clouds observed on 22-25 January 2016 and 28 February-3 March 2017 were not in the immediate vicinity, relatively fast-moving gravity waves caused by deep convection could spread the deep convective temperature signals over
large regions in short amounts of time (Holloway and Neelin, 2007). The temperature anomalies in Figure 7 are much larger than those reported by Paulik and Birner for temperature profiles around the time (± 6 hours) and location of deep convection (within 1000 km).  However, we are studying deep convective temperature anomalies associated with two





individual events while their deep convective temperature signal was estimated using 4 years of COSMIC data. Therefore, their estimates correspond to an average deep convective temperature signal; such a signal is likely larger when considering

larger/more organized convective events such as tropical storms.

**4.5 CFH and MLS comparisons**

The CFH measurements analyzed in this study are further compared to coincident MLS profiles. The match criteria used are ±18h, ±500 km North-South distance (around ±5° latitude), ±1000 km East-West distance (around ±10° longitude). The same match criteria are used in Davis et al. (2016). 10 and 6 matched MLS profiles are found for 25 January 2016 and 3

March 2017 respectively.  On 25 January 2016, distances between the Maïdo Observatory and the matched MLS profiles range from 259 to 946 km, with a mean distance of 554 km. The mean time difference for all matched profiles is 9 h.  On 3 March 2017, the 6 matched MLS profiles are closer to the Maïdo Observatory with a mean distance of 309 km and are east of the island (2 north of the island, 2 at the latitude of the island and 2 south of the island). However, a larger mean time difference of 14 h is observed for the matched MLS profiles.

To compare the high-resolution CFH water vapor profile to the MLS satellite data, we smooth the high resolution sonde measurements to match the resolution of the satellite profiles using the MLS vertical averaging kernels, following the procedure described in Read et al. (2007) and Davis et al. (2016). The procedure for applying the MLS averaging kernels to a CFH profile requires an a priori profile as input; this is the same a priori profile used in the MLS retrieval. Figure 9 shows the matched MLS profiles and the CFH profiles convolved with the MLS averaging kernels. The matched MLS profiles on

both dates illustrate how water vapor is more variable in the upper troposphere between 316 and ~147 hPa. Above 100 hPa, MLS collocated profiles show less variability and are relatively close to the mean MLS profile. The lower part of the tropopause layer from 147 hPa to the cold point tropopause (green dashed line on Figure 9) is a transition region where water vapor mixing ratios become lower but could still be influenced by deep convective outflow. The application of the averaging kernel to the CFH profiles smooths the fine-scale structures observed in the CFH profiles on Figure 5 but still captures the

deep layers of moist air in the upper troposphere between 261 and 147 hPa.  To facilitate the comparison of CFH and MLS water vapor profiles in the upper troposphere and stratosphere where water vapor mixing ratios decrease by 3 orders of magnitude, we compute a mean percent difference of the MLS collocated profiles to the CFH and MLS data (i.e., percent difference = (MLS - CFH)/((CFH + MLS)/2)x100). The same definition is used in Davis et al. (2016) and ensures that the distribution of percent difference at each pressure level is not skewed toward positive values larger than 100% (since water

vapor values are constrained to be positive). In addition, this facilitates comparison with the study of Davis et al. (2016) that established a comparison between the 2004-2015 MLS water vapor data record and both routine monitoring and field campaign frost point hygrometer balloon soundings at various stations around the world.



The mean percent difference between the collocated MLS profiles and CFH convolved profile is shown on the right panel of Figure 9. In the upper troposphere between 316 and 178 hPa, MLS profiles tend to be drier than the CFH measurements by 445 23 ± 11% on 25 January 2016 and 40 ± 28% on 3 March 2017. The MLS dry bias found in the upper troposphere is in agreement with previous studies (Vömel et al., 2007b; Davis et al., 2016; Yan et al., 2016).

The larger upper tropospheric dry bias on 3 March 2017 can be explained by the fact that the collocated MLS profiles do not capture the peak of water vapor which is observed in the convolved CFH profile at 215 hPa. The wet layer between 261 and 147 hPa was observed between 12 and 15 km in the high-resolution CFH profile in Figure 5. By using Lagrangian 450 trajectories, we will later show that this peak is associated with the convective outflow of tropical cyclone Enawo. The vertical resolution of MLS profiles is of the order of 2 to 3 km for water vapor and thus the 3-km deep wet layer observed on 3 March 2017 will not be well captured by the MLS profiles as opposed to the CFH. On 4 March 2017, the two collocated MLS profiles found north of the island (at 10:21 and 10:22 UTC) have larger water vapor mixing ratios between 316 and 178 hPa than the 2 collocated MLS profiles found south of the island at ~ 10:20 UTC. The two collocated MLS profiles 455 north of the island were closer to tropical storm Enawo with a distance of ~ 660 km from the storm center located at 13.54°S, 56.87°E on March 4 at 9 UTC. In contrast, the two collocated MLS profiles south of the island were ~ 1075 km from the storm center. At 261 hPa, there is a water vapor mixing ratio difference of ~230 ppmv between the northernmost and southernmost collocated MLS profiles on March 4. This would be coherent with a sharp south-north gradient in water vapor in the upper troposphere due to the proximity of TC Enawo. MLS has a 500 km along-track resolution and could have 460 measured the south-north gradient in water vapor. The mean percent difference at 316 and 261 hPa decreases by 60% if we remove the two northernmost collocated MLS profiles on March 4 (not shown).

Below the tropopause (100-147 hPa), collocated MLS profiles are larger by 22 ± 8% on 25 January 2016. There is a sharp gradient between 15 and 18 km in the CFH profile on 25 January 2016 shown on Figure 5. The MLS vertical resolution of ~3 km in the TTL implies that sharp gradients observed on 25 January 2016 will most likely not be reproduced by MLS and 465 as a result the retrievals may be wet biased. In addition, the moist bias of MLS between 147 and 100 hPa may reflect weather-related horizontal gradients in water vapor in the upper troposphere and near the tropopause. If we only compare the convolved CFH profile to the 5 collocated MLS profiles measured on 25 January 2016 at ~ 21:40 UTC east of Réunion (between 55°E and 60°E), the differences at 147, 121 and 100 hPa become less important. The mean percent differences decrease to 2% (+1.2pmmv), 26% (+1.4 pmmv) and 3% (+0.1 pmmv) at 147 hPa, 121 hPa and 100 hPa respectively. The 470 MLS profiles measured on 26 January 2016 beyond 60°E were most likely more influenced by tropical storm Corentin in the upper troposphere and near the tropopause. The mean wet biases of +12.5% and +9.5% at 100 hPa observed on 25 January 2016 and 3 March 2017 respectively are in agreement with wet biases reported in Davis et al. (2016) and Hurst et al. (2014).





On 25 January 2016, the mean MLS water vapor profile agrees well with the convolved CFH profile over the entire lower tropical stratosphere above 68 hPa. The mean percent difference between 68 and 22 hPa is +6.3 ± 1.6% (+0.3 ppmv) and lies
within the previously published uncertainty bounds of the instrument (Hurst et al., 2014; Vömel et al., 2007a; Davis et al., 2016; Yan et al., 2016).

However, on 3 March 2017, large differences of up to +30% (~1 ppmv) are observed in the stratosphere at 68 and 32 hPa. The mean percent difference in the stratosphere above 83 hPa is 20% ± 5.5% (+0.8 ppmv). It is not clear why there are larger differences in the stratosphere on 3 March 2017. Both CFH instruments launched on 25 January 25 2016 and 3 March 2017
were prepared by the same operator and calibrated using the same recommended procedure. During these two flights, the CFH data streams were transmitted to receiving equipment on the ground through the Intermet radiosonde. From an instrumental standpoint, there is nothing that might explain a CFH dry bias on 3 March 2017 compared to 25 January 2016. Unfortunately, the CFH sondes are not recovered on the island after each flight as they land in the ocean and thus it was not possible to examine in more details the instrument after the flight on 3 March 2017. To our knowledge, the CFH instrument
on that night has measured as well as it could in the stratosphere. Even though the CFH instrument launched on 3 March 2017 had a dry bias of 1 ppmv in the stratosphere, such bias does not affect the results of this paper found for TC Enawo.

## 5 FLEXPART Lagrangian analysis

The convective origin of air masses sampled in the upper troposphere and near the tropopause during the passage of TS Corentin and TC Enawo is evaluated using the FLEXPART Lagrangian model. Figures 10 presents the origins of air masses
sampled within the 14-15 km (~147-121 hPa) and 17-18 km (100-83 hPa) layers, altitudes that correspond to RHi peaks on Figure 6, on January 25 2016 above the Maïdo Observatory. The origins and pathways of these air masses were examined by computing 10-day FLEXPART back trajectories. On Figure 10, the origins of air masses measured in the upper troposphere (14-15 km, 147-121 hPa) and near the tropopause (17-18 km, 100-83 hPa) are shown for one day and two days prior to the launch. The position of each air mass is depicted by 10000 dots color coded by their altitude and is overlaid over
METEOSAT 7 infrared images valid at the time of the back trajectories. For example, trajectories that were originally in the lower troposphere (below 5 km) and middle troposphere (between 5 and 10 km) one/two days before are indicated by orange and brown dots respectively. In other words, these air masses were transported from the troposphere to the upper-troposphere (14-15 km)/tropopause region (17-18 km) in one or two days before being sampled by the CFH instrument on 25 January 2016 around 18:30 UTC above the Maïdo Observatory. The air mass fractions for different altitude ranges are
also indicated at the bottom of Figure 10. Variations in the air mass fractions over time (e.g. from the lower troposphere below 5 km) can be interpreted in terms of changes in the vertical transport due to convection over the SWIO.



The ability of FLEXPART to represent isolated deep convective cells is limited, due to the $0.15^O$x$0.15^O$ spatial resolution of the ECMWF operational fields. At that resolution, isolated deep convective cells are not fully resolved in the ECMWF vertical wind field, and their updraft intensity and the altitude of the level of neutral buoyancy could be underestimated.

However, the vertical transport of convective cells organised at mesoscale such as convection in tropical cyclones that cover several degrees in longitude and latitude are better resolved by the $0.15^O$x$0.15^O$ ECMWF meteorological fields. Hence the FLEXPART backtrajectories driven by the ECMWF operational wind field give a qualitative sense of convective origins of vertical layers measured at Maïdo in relation to tropical cyclones.

According to FLEXPART, the 14-15 km layer measured above the Maïdo Observatory on 25 January 2016 ~18:30 UTC has

two different origins. A day before, 69% of this air mass was below 10 km (with ~29% below 5 km) and ~1000 km northeast of Réunion Island in a region with convective clouds with cold brightness temperatures less than 220 K (~12 km). Therefore, we can infer that the majority of the 14-15 km air mass was lifted by convection associated with TS Corentin a day prior to the launch. These trajectories are rather spread in the lower troposphere, suggesting that they experienced turbulent mixing and changes in wind direction in the lower troposphere. The rest of the trajectories are located higher in altitude, in the 10-15

and 15-17 km altitude ranges. They are also located above convective clouds, but are less scattered than the trajectories in the lower troposphere, suggesting that these trajectories were less mixed with the surrounding upper troposphere.

2 days before (Figure 10b), 80% of the 14-15 km layer originated from the lower and middle troposphere (54% within the 0-5 km layer, 26% within the 5-10 km layer) over the northeastern convective region of TS Corentin, and 20% from the upper troposphere and near tropopause region (13% within 10-15 km, 6% within 15-17 km) above TS Corentin. The upper

tropospheric branch had an anticlockwise rotation with an origin near TS Corentin, in agreement with the upper divergence associated with TS Corentin. Hence, most of the 14-15 km air mass was located either in the lower troposphere or near the top of convective clouds 2 days before.

The 17-18 km layer measured at Maido on 25 January 2016 stayed in the upper troposphere and near the tropopause a day before before reaching Réunion Island. The trajectories followed an anticlockwise rotation associated with Corentin's

dynamics. No trajectories that originate in the lower troposphere were found. On 24 January at 17 UTC (2 days before the launch), the trajectories were located ~250 km north of the center of TS Corentin. Note that TS Corentin reached its peak intensity on 23 January 2016 at 06 UTC (pressure at the center of 975 hPa, ten-minute maximum sustained winds of 111 km/h). Hence, according to FLEXPART backtrajectories and the METEOSAT 7 infrared images, the origin of the 17-18 km layer was traced back to the active convective regions of TS Corentin and its upper divergence dynamics, but no trajectories

originated from the lower troposphere. However, due to the 0.15° spatial resolution of the ECMWF winds used to drive FLEXPART, the vertical updrafts of the deepest convective clouds that may reach the tropopause region/lower stratosphere





may not be well represented in FLEXPART.

Figure 11 is similar to Figure 10 but for backtrajectories associated with the launch on 3 March 2017. Most of the 14-15km layer measured on 3 March 2017 at 18:42UTC was lifted by convection on 2 March 2017 at 17:00 UTC 800 km north of the
island one to two days before (Figure 11a&b). A day before (Figure 11a), the backtrajectories indicate that a large fraction (68.6%) of the 14-15 km airmass is from the lower troposphere (below 10 km) over a convective region associated with TC Enawo. 2 days before reaching Réunion Island (Figure 11b), the trajectories were dispersed in the lower troposphere around the forming storm as Enawo was in the early stage of its formation at that time (tropical depression).

The FLEXPART backtrajectories for the 17-18 km air mass measured above the Maïdo Observatory on 3 March 2017 at
18:52 UTC stayed in the upper troposphere one and two days before the launch (Figure 11c&d). The trajectories were confined to the same latitude band east and west of Réunion Island in a clear sky region, away from convective clouds. It shows that air masses near the tropopause above Réunion Island on 3 March 2017 were most likely not affected by Enawo at this stage of its development as Enawo was still intensifying.

In a nutshell, the FLEXPART backtrajectories clearly identify a convective origin for the 14-15 km layer sampled on 25
January 2016 and 3 March 2017 associated with TS Corentin and TC Enawo tropical cyclones. The convective transport from the lower troposphere to the upper troposphere occurred roughly a day before each launch at 1100 km and 800 km respectively away from the island. As for the tropopause region over Réunion Island on 25 January 2016, FLEXPART backtrajectories suggest that the air masses were embedded in TS Corentin upper divergence dynamics over a region where convection was active. Deep convective clouds within TS Corentin may have reached the tropopause region (83-100 hPa) on
23 January 2016 when the storm was at its peak intensity and may have influenced the water vapor content near the tropopopause (~83 hPa on Figures 9&12). However due to the model resolution, the backtrajectories initialized for the 17-18 km layer do not show a lower tropospheric origin for these air masses even though they were located above the convective outflow of TS Corentin on the previous days. On 3 March 2017, the tropopause region measured by the CFH sounding was not affected by deep convection associated with Enawo according to the model, at least not at the time of the observation. At
that time TC Enawo was still intensifying and the deepest convective cloud developed later after 4 March 2017.

## 6. Discussion

To further assess the impact of TS Corentin and TC Enawo on the UTLS water vapor content, we compare the convolved CFH profiles to a monthly climatological MLS water vapor profile as there are no long-term stratospheric water vapor measurements at Réunion Island. For each year between 2004 and 2017, MLS water vapor profiles within ±5° latitude and
±10° longitude of Réunion Island and over a period of 15 days surrounding the launch date, i.e. 10 January-9 February for 25



January 2016 and 16 February-18 March 18 for 3 March 2017, are used to define a monthly climatological water vapor profile. The monthly climatological MLS water vapor profiles and CFH convolved profiles are shown on Figure 12. Both monthly climatological water vapor profiles have comparable minimum water vapor mixing ratio at 83 hPa (3.5 ± 0.6 ppmv and 3.3 ± 0.5 ppmv for the January and March climatologies respectively). In the upper troposphere (316-178 hPa) the

climatologies have mean values of 277.6 ± 269.2 ppmv and 266.1 ± 253.2 ppmv for January and March respectively. High variability in the UT is consistent with deep convection being more active during austral summer. Higher UT water vapor content in January relative to March is in agreement with the fact that the austral summer season reaches its peak in January/February. Both January and March climatologies have comparable TTL (147-68 hPa) water vapor content (5.3 ± 1.8 ppmv and 5.1 ± 1.7 ppmv for January and March respectively). The climatological mean stratospheric (56-22 hPa) value is

4.2 ± 1.3 ppmv for both months.

Relative water vapor differences are defined with respect to the monthly climatological profile (i.e., relative difference = (CFH - MLS Climatology)/MLS Climatology x100) and are displayed on the bottom panels of Figure 12. In addition to the CFH convolved profile, we also compared the mean of MLS coincident profiles to the MLS monthly climatological profile for 25 January 2016 and 3 March 2017. On 25 January 2016, the 5 MLS profiles located east of Réunion Island between 55

and 60°E are used to compute the mean MLS profile for that day.

On both dates, the CFH convolved profiles and the mean of MLS coincident profiles are drier than the MLS monthly averages at 316 and 261 hPa with relative differences ranging from -10 % to -70 %. The mean relative difference with the climatology for these two pressure levels is ~ -20% for the CFH convolved profiles and the means of MLS coincident profiles respectively.

On 25 January 2016, the mean of MLS coincident profiles and the CFH convolved profile show a peak of ~ 30% in the relative difference with the MLS climatology but the pressure level of this peak differs in the two profiles with a peak at 178 hPa for the CFH convolved profile and 147 hPa for the mean of coincident MLS profiles. Overall, the region of moistening in the upper troposphere is broader by ~ 1.5 km in the mean of MLS coincident profiles compared to the CFH convolved profile. MLS water vapor data are retrieved on a grid having 12 levels per decade change in pressure, corresponding to ~1.3

km spacing from 316 hPa to 22 hPa but the vertical resolution is 2 to 3 km in the upper troposphere. This could explain the somehow broader region moistening of the UT observed on 25 January 2016. FLEXPART backtrajectories initialized on 25 January 2016 for the 14-15 km layer (121-147 hPa) indicate a convective source for this layer. FLEXPART backtrajectories initialized around 178 hPa (not shown) on 25 January 2016 present a similar behavior to the 14-15km layer trajectories shown on Figure 10. This confirms that the positive water vapor anomalies observed at 147 and 178 hPa on 25 January 2016

for the CFH and the mean of MLS coincident profiles (bottom left panel of Figure 12) are associated with the convective



outflow of TS Corentin.

On 3 March 2017, the hydration of the upper troposphere between 215 and 121 hPa is much more pronounced in the CFH convolved profile with a mean 121-215 hPa relative difference of 100% and a peak of ~200% at 178 hPa. For the mean of MLS coincident profiles, the moistening is not as large with a mean 121-215 hPa relative difference of 20% and a peak of

~42% at 147 hPa. Several factors could explain why the water vapor enhancement is much larger in the CFH profile. First, the 3-km deep wet layer observed on March 2017 in the CFH profile will not be well captured by MLS with a 2-3 km vertical resolution in the upper troposphere. In addition, the CFH launch on 3 March 2017 at 18 UTC was planned using FLEXPART Lagrangian trajectory analysis and satellite images in the days prior to the launch to sample the convective detrainment of TC Enawo. Therefore, the planning of the CFH launch on 3 March 2017 was optimal to sample moist air

from convective detrainment and an average of 6 MLS coincident profiles over a larger region/time window could be an underestimate of the storm related moistening. The CFH measurements indicate a moistening of the 121-215 hPa layer by 100 % or ~ 30 ppmv. FLEXPART backtrajectories initialized around that layer (not shown) present a similar behavior to the 14-15km (121-147 hPa) layer trajectories shown on Figure 11. Ray and Rosenlof (2007) used measurements from AIRS to assess the impact of tropical cyclones in the Atlantic and Pacific basins on the amount of water vapor in the tropical UT.

They showed that tropical cyclones can hydrate a deep layer of the surrounding upper troposphere by ~30-50 ppmv or more within 500 km of the eye compared to the surrounding average water vapor mixing ratios (cf. Figure 3 of Ray and Rosenlof, 2007). They also looked at the evolution of UT water vapor changes as a function of the storm intensity as measured by the peak wind speed (cf. Figure 5 of Ray and Rosenlof, 2007). In both the Atlantic and western Pacific basins, the average water vapor at 223 hPa around the storm center steadily increased from 4 to 5 days prior to peak cyclone intensity to 2 days

following peak cyclone intensity. The average water vapor enhancement in the two ocean basins was from 5 to 20 ppmv with an increase as high as 30-40 ppmv for some cyclones in the western Pacific. The CFH launch on 3 March 2017, 18 UTC occurred 3.5 days before Enawo reached its peak intensity on 7 March at 06 UTC (pressure at the center of 932 hPa, ten-minute maximum sustained winds of 204 km hr$^{-1}$) and the storm center was ~ 700 km away from the island. Thus, deep convection associated with TC Enawo may have caused the strong increase in UT water vapor observed on 3 March 2017.

Ongoing work with MLS data to apply the methodology of Ray and Rosenlof (2007) to assess hydration of the UTLS by tropical cyclones for the 2004-2017 cyclone seasons in the southwest Indian Ocean is under way. This will be the focus of a future study but preliminary results indicate water vapor differences of 35% to 48% at between 178 and 261 hPa for categories 2 to 4 hurricanes on the Saffir-Simpson scale. Ray and Rosenlof (2007) indicated that tropical cyclones hydrate a deep layer of the UT in the vicinity of the cyclones by up to 50% above monthly mean water vapor mixing ratios. Therefore,

our estimate of UT water vapor increases of 20 to 100% using CFH&MLS data for TS Corentin (Category 1 hurricane at its peak intensity) and TC Enawo (Category 4 hurricane at its peak intensity) are in broad agreement with our estimates based





on the 2004-2017 MLS data and the study of Ray and Rosenlof (2007).

On 3 March 2017, at 121 hPa (~15.4 km) both MLS and CFH data are above the climatological monthly mean values by 6% and 9% respectively (thus a moistening of the 121 hPa layer by 0.2 to 0.3 ppmv). In Section 5, FLEXPART Lagrangian
backtrajectories indicated a convective origin for this layer. Ueyama et al. (2018) used Lagrangian trajectories to assess the convective influence on the 100 hPa water vapor during boreal summer 2007 with a focus on the Asian monsoon region. They established that over the tropics (10°S-50°SN) convection moistens the 100 hPa level by ~0.6 ppmv and that the Asian monsoon (over the region 0-40°N, 40-140°E) was responsible for ~0.3 ppmv of this increase. On 3 March 2017, the tropopause over Réunion Island was located at 113.5 hPa and the moistening of ~0.2-0.3 ppmv observed at 121 hPa due to
TC Enawo is in broad agreement with the convective moistening found by Ueyama et al. (2018) due to the Asian monsoon convection as both convection associated with TC Enawo and the Asian monsoon corresponds to convection organised at mesoscale/large scale.

At 100 hPa, both MLS and CFH data are below the climatological monthly mean values. On 25 January 2016, MLS and CFH data are 20% below the climatological values (-0.7 ppmv) and the relative difference is less important on 3 March 2017
(5 and 13%, -0.2 and -0.5 ppmv for MLS and CFH data respectively). This would be coherent with the near tropopause cooling observed on Figure 7 and the presence of deep convection around Réunion Island on Figure 8. In addition, TTL cirrus clouds were observed north of the island on both dates (Figure 6). Convectively generated or in-situ cirrus clouds in the TTL can dehydrate the tropopause region. Jensen et al. (1996) showed that ice clouds formed by large-scale vertical motions can result in depletion of water vapor mixing ratio by about 0.4 ppmv. Chae et al. (2011) investigated temperature
and water vapor changes due to clouds in the TTL using MLS, CALIPSO and CloudSat datasets. They noted that generally clouds humidify the environment near 16 km (~100 hPa) or lower but dehydrate the TTL above 16 km.

On 25 January 2016, CFH and MLS data are 10% (+0.4 ppmv) and 17% (+0.7 ppmv) moister than the climatological values at 68 hPa, above the tropopause. Observational and modeling studies have indicated that overshooting convection can moisten the lower stratosphere by injecting water vapor or ice crystals directly above the overshooting clouds (e.g.
Danielsen, 1993; Corti et al., 2008; Dauhut et al., 2015; Frey et al., 2015; Allison et al., 2018). In our case, the observation on 25 January 2016 was not made close to the deepest convective clouds that were ~1000 km north of the island (Figure 8), but was downwind of TS Corentin (See section 5), as shown by the FLEXPART analysis (Figure 10). However, FLEXPART backtrajectories indicate that the air masses at 68h Pa originate from the South East Indian Ocean in the 20°S-30°S latitude band (not shown) where the MLS water vapor anomaly for January 2016 is around 0.5 ppmv most likely due to
the impact the 2016 strong El Niño event. The QBO easterlies can be observed at 70 hPa. Hence, the positive anomaly against the climatological value can be explained by horizontal advection from the South East Indian Ocean toward Reunion





island.

It is difficult to conclude whether TC Enawo had a direct impact on water vapor in the lower stratosphere by using only the CFH observation on 3 March 2017. The FLEXPART analysis indicated that the CFH sounding did not sample the lower
stratosphere downwind of Enawo.

Ongoing work with the mesoscale model Meso-NH at a 2-km resolution for TC Enawo for the period 2-7 March 2017 indicates that deep convective clouds within 500 km of the cyclone eye can inject ice crystals and moisten the lower stratosphere, resulting in an average anomaly of ~2ppmv within 500 km of the tropical cyclone eye. The strongest humidification in the lower stratosphere (17-19 km; ~88-66 hPa) was found after March 4 when the storm stalled over the
ocean (while intensifying) and after March 6 when it reached its peak intensity. Thus, the CFH observation on 3 March 2017 was made before TC Enawo had influenced the lower stratosphere above 100 hPa. This is further confirmed by the fact CALIOP did not have a lower stratospheric signal on Figure 6.

Tropical cyclones are unique among tropical convective systems in that they persist for many days and thus could affect the UTLS more than other mesoscale convective systems. Clouds in tropical cyclones often reach to and sometimes beyond the
tropopause (e.g., Romps and Kuang 2009). Allison et al. (2018) have investigated the vertical transport of water vapor by the 2013 tropical cyclone Ingrid in the North Atlantic. Results of their high-resolution numerical simulations indicated that hydration occurred between 17.5 and 21 km (83 to 56 hPa) due to the injection of ice crystals. As the exact role of deep convection, and tropical cyclones in particular, in hydrating the lower stratosphere is still under debate, additional TTL observations of water vapor and modeling work are needed to quantify the overall impact of convection on TTL and LS
water vapor. High-resolution (2 km) numerical simulations of TC Enawo for the period 2-7 March 2017 are underway to gain a closer look at the effect of TC convection on TTL temperature and water vapor. This work will be the subject of a subsequent study.

**7 Summary**

Two balloon launches from the Maido Observatory were specifically planned using the FLEXPART Lagrangian model and
METEOSAT 7 infrared images to sample the convective outflow from Tropical Storm Corentin on 25 January 2016 and Tropical Cyclone Enawo on 3 March 2017. Balloon-borne measurements of CFH water vapor, ozone and iMET temperature and water vapor lidar measurements, showed that both storms humidified the TTL, with $RH_{ice}$ values exceeding 50% for TS Corentin and 90% for TC Enawo in the upper troposphere. Comparing the two CFH profiles to the climatological monthly mean MLS water vapor profiles, positive anomalies of water vapor were identified (10 ppmv for TS Corentin and 30 ppmv
for TC Enawo) between 12 and 15 km in altitude (247 to 121 hPa) for both storms. According to the FLEXPART



backtrajectories and METEOSAT 7 infrared images, those air masses originated from convectively active regions of TS Corentin and TC Enawo and were lifted from the lower troposphere to the upper troposphere around one day before the planned balloon launches. In addition, the CALIOP satellite measurements indicated cirrus clouds north of Réunion Island for the same altitude range for both storms.

According to the CFH profile on 25 January 2016 and MLS climatology, air masses measured near the tropopause were anomalously dry around 100 hPa and anomalously wet around 68 hPa in the lower stratosphere. FLEXPART backtrajectories were used to find the origin of these layers that could be traced back to TS Corentin upper-tropospheric divergent flow and active convective regions. Deep convective clouds and cirrus clouds may have dehydrated the region around 100hPa. According to FLEXPART backtrajectories, the positive anomaly at 68hPa can be explained by a horizontal

transport from the South East Indian Ocean. The South East Indian Ocean had a positive water vapor anomaly of ~0.5ppmv in January 2016 most likely due to the strong 2016 El Niño event (Avery et al., 2017).

On the contrary, no water vapor anomaly was found near or above the tropopause on 3 March 2017 as the tropopause region was not downwind of TC Enawo. According to FLEXPART backtrajectories, those air masses stayed away from the upper-tropospheric dynamics of TC Enawo and its convective active regions. Hence the tropopause region on 3 March 2017 was

not affected by Enawo, at least not at the time of the balloon launch and at this stage of Enawo's development.

This study showed the impact of two tropical cyclones on the humidification of the TTL. It also demonstrates the need to develop balloon borne high precision observations in regions where TTL in-situ observations are sparse, such as the tropics and the SWIO in particular. High-resolution accurate observations of water vapor are needed to document the impact of tropical cyclones and deep convection in general on the TTL. The impact of tropical cyclones on the TTL water vapor

budget will be analyzed in a more quantitative way using MLS data and tropical cyclones best tracks from 2004 to 2017 in a subsequent paper. In addition, the impact of deep convection and overshooting clouds within TC Enawo on the water vapor budget of the TTL will be analyzed using high-resolution (2 km) mesoscale simulation of TC Enawo.

*Data availability.* MLS water vapor data used in this study are available at https://mls.jpl.nasa.gov/ and CALIPSO L1B lidar data are available at https://eosweb.larc.nasa.gov/project/calipso/lidar_l1b_profile_table. The NDACC/SHADOZ ozone

measurements for Réunion Island are available at https://tropo.gsfc.nasa.gov/shadoz/Reunion.html. The SWOOSH dataset is available at https://data.nodc.noaa.gov/cgi-bin/iso?id=gov.noaa.ncdc:C00958. The CFH and lidar water vapor data are available from the authors (SE, VD, PK) upon request. The FLEXPART Lagrangian trajectories can be requested from the corresponding author Stephanie Evan (stephanie.evan@univ-reunion.fr).



*Author contributions.* All authors contributed to the paper. SE wrote the manuscript with contributions from JB, KR, SD, DH, FP, JMM, VD, GP, HV, PK, JPC. SE, JB, FP, JMM, DH, JPC, VD, GP and HV performed the CFH/Ozone/Lidar measurements. HV processed the CFH data. SE and JB performed the FLEXPART simulations. SM provided the SWOOSH dataset. All authors revised the manuscript draft.

*Competing interests.* The authors declare that they have no conflict of interest.

*Acknowledgments*

We thank the Aura Science Team for the MLS data (https://mls.jpl.nasa.gov/) and the CALIPSO science team for the L1B lidar data (https://eosweb.larc.nasa.gov/project/calipso/lidar_l1b_profile_table).

OPAR (Observatoire de Physique de l'Atmosphère à La Réunion, including Maïdo Observatory) is part of OSU-R (Observatoire des Sciences de l'Univers à La Réunion) which is being funded by Université de la Réunion, CNRS-INSU,

Météo-France, and the french research infrastructure ACTRIS-France (Aerosols, Clouds and Trace gases Research Infrastructure). OPAR's water vapor lidar and ozone radiosounding belong to the international network NDACC (Network for the Detection of Atmospheric Composition Change). This work was supported by the French LEFE CNRS-INSU Program (VAPEURDO).

S. Evan thanks Susanne Koerner (DWD/GRUAN Leadcentre, Germany) for her training on the CFH instrument.

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








**Table 1:** CPT properties (temperature and height) from the radiosonde launches on 25 January2016/3 March 2017 and NDACC/SHADOZ seasonal mean (December-March) CPT properties (for the period 1997-2017).

| | Observations | CPT T (K) | CPT altitude (km) |
|---|---|---|---|
| mean SHADOZ Dec-March (1997-2017) | 200 | 193.90 (±2.26) | 17.31 (±0.71) |
| Profile on 25 January 2016 | 1 | 192.64 | 17.30 |
| Profile on 3 March 2017 | 1 | 194.58 | 16.10 |






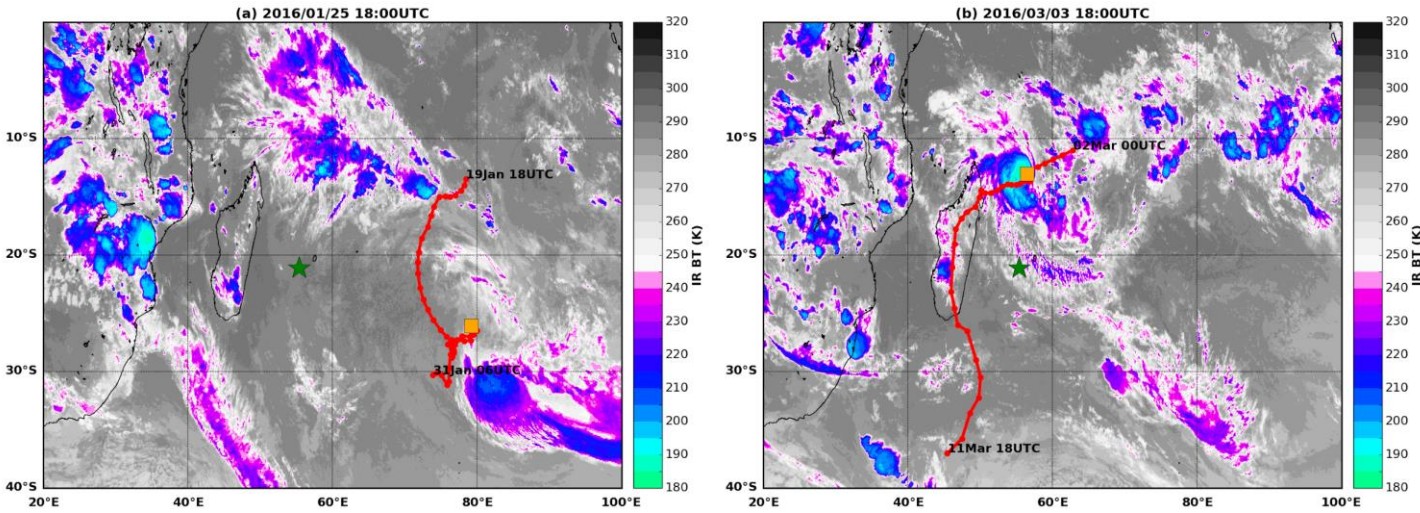

**Figure 1**: Infrared (10.8 µm) brightness temperature (K) observed by METEOSAT-7 at the time of the CFH launch for a) 25 January 2016 at 18UTC and b) 3 March 2017 at 18UTC. The red lines correspond to the best tracks of tropical cyclones Corentin (19-31 January 2016) and Enawo (02-11 March 2017). The orange squares indicate the positions of the TC centers (defined as the minimum pressure in the Météo-France best track data) at the time of the satellite observation. The green stars indicate the position of the Maïdo Observatory on Réunion Island (21.08ºS, 55.38ºE).





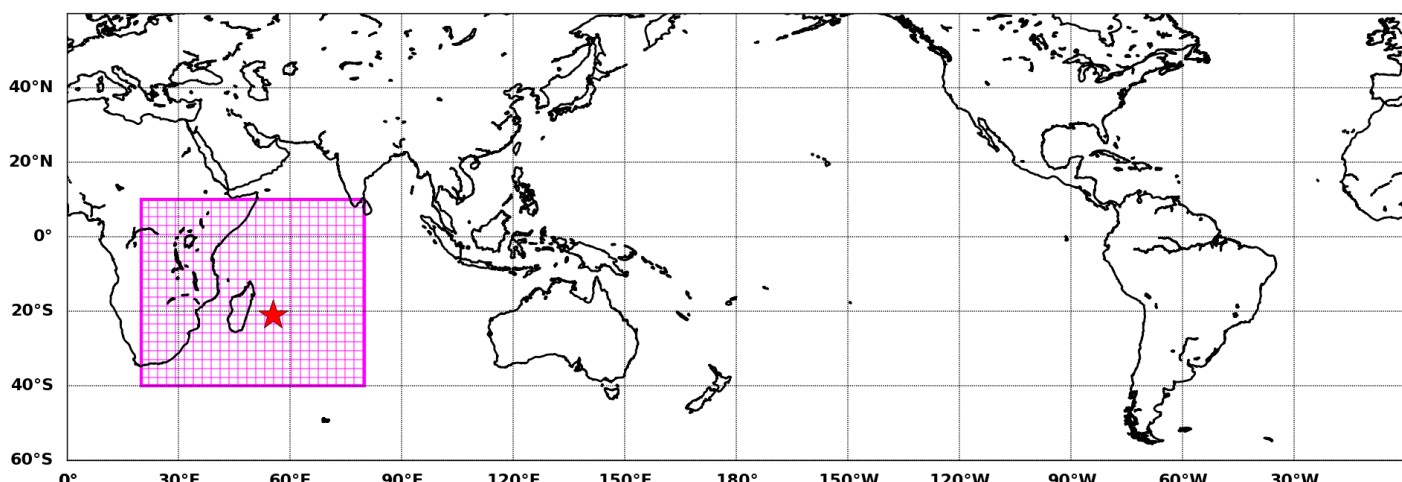

**Figure 2**: Map showing the FLEXPART mother domain at 0.50°x0.50° and higher resolution nest domain at 0.15°x0.15°
925covering the SWIO. The red star indicates the position of the Maïdo Observatory on Réunion Island (21.08ºS, 55.38ºE).





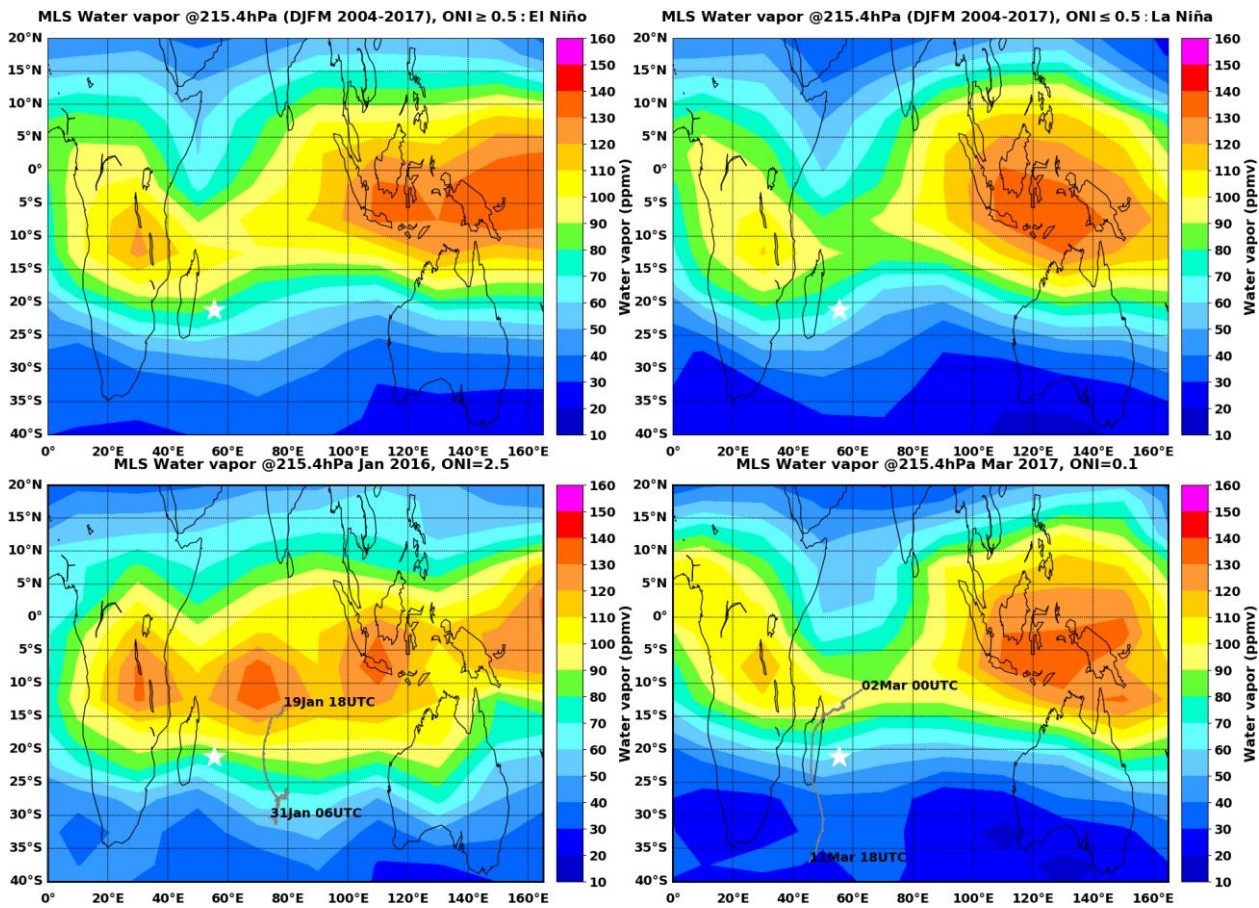

930

**Figure 3**: MLS water vapor mixing ratio (ppmv) gridded in the SWOOSH data set at 215 hPa averaged over December-March 2004-2017 for El Niño conditions (ONI ≥ 0.5, upper left) and for La Niña conditions (ONI ≤ 0.5, upper right). SWOOSH water vapor mixing ratio (ppmv) at 215 hPa for January 2016 (lower left) and for March 2017 (lower right). The gray lines correspond to the best tracks of tropical cyclones Corentin (19-31 January 2016) and Enawo (02-11 March

935    2017).





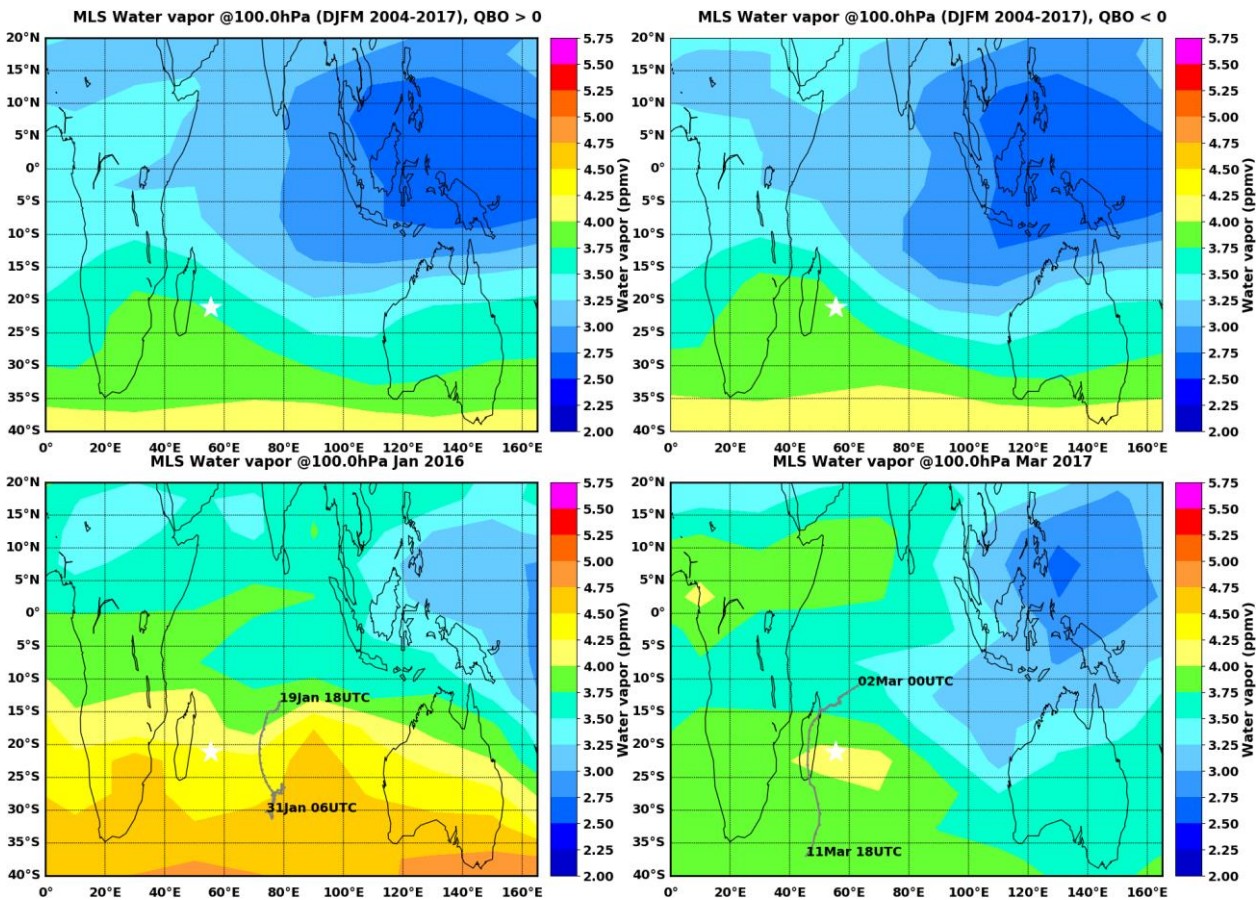

**Figure 4**: MLS water vapor mixing ratio (ppmv) gridded in the SWOOSH data set at 100 hPa averaged over December-March 2004-2017 for QBO westerly shear phase conditions (QBO > 0, upper right) and QBO easterly shear phase conditions (QBO <0, upper left). SWOOSH water vapor mixing ratio (ppmv) at 100 hPa for January 2016 (lower left) and for March 2017 (lower right). The gray lines correspond to the best tracks of tropical cyclones Corentin (19-31 January 2016) and Enawo (02-11 March 2017).





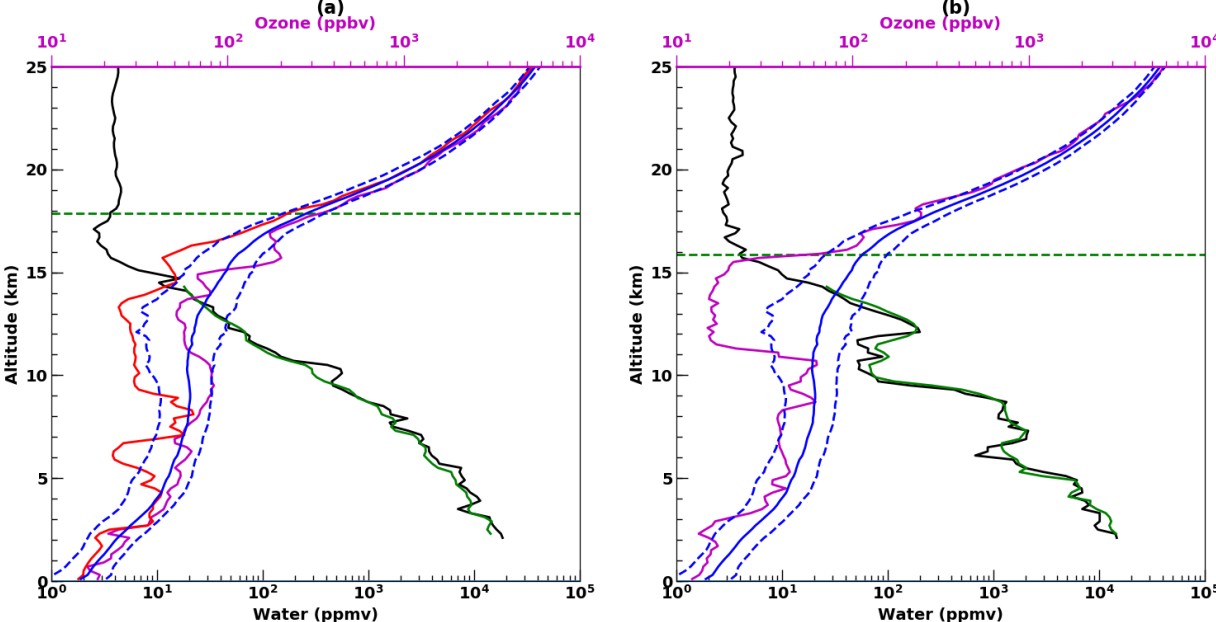

**Figure 5**. Vertical profiles of: a) CFH and lidar water vapor profiles (ppmv) on 25 January 2016 (black and green line respectively), NDACC/SHADOZ ozone profiles on 18 January 2016 (purple line) and 4 February 2016 (red line); b) CFH and lidar water vapor profiles (black and green line respectively) and NDACC/SHADOZ ozone profile (purple line) on 3 March 2017. The location of the cold point tropopause is indicated by the dashed green line. Also shown on each plot is the 1998-2017 climatological mean ozone profile (blue line) and the standard deviation of the climatology (dashed blue line) for DJFM.





**Figure 6**: Top and bottom left: vertical profiles of temperature and relative humidity with respect to ice (black and blue line respectively) measured on 25 January 2016 at 17:52 UTC and 3 March 2017 at 18:00 UTC. Top and bottom right: Latitude-altitude distribution of CALIOP backscattering ratio at 532 nm along CALIOP track near Réunion Island on 25 January 2016 (top right) and 3 March 2017 (bottom right). The mean longitude difference between the CFH profile and the CALIOP track is 2.4° on 25 January 2016 and 5.3° on 3 March 2017. The red curve on each CALIOP plot corresponds to the tropopause height provided by the GEOS 5 global model data available in the CALIPSO Level 1 data files. The latitude of the Maïdo Observatory is indicated by the black star on each CALIOP plot.

**Figure 7**: Upper right panel: Temperature profile measured by the iMet radiosonde at the Maïdo Observatory on 25 January 25 at 17:52 UTC (red line), NDACC/SHADOZ climatological-mean summertime (December-March) profile of temperature (black line) and standard deviation (grey shading). Upper left panel: Temperature profile measured by the iMet radiosonde at the Maïdo Observatory on 3 March 2017 at 18:00 UTC (red line), NDACC/SHADOZ climatological-mean summertime (December-March) profile of temperature (black line) and standard deviation (grey shading). The bottom panels correspond to the temperature difference between the iMet temperature profile and the NDACC/SHADOZ climatological-mean.

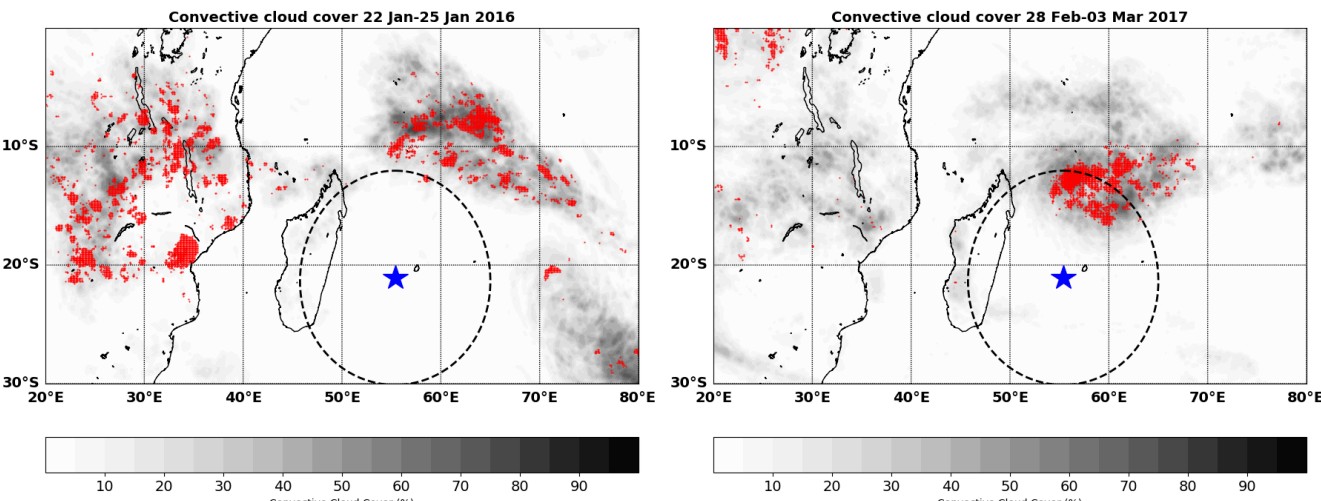

**Figure 8:** Maps of convective cloud cover (gray shading) computed using 3-hourly data of METEOSAT 7 infrared brightness temperature at 5 km resolution for 22-25 January 2016 (left) and 28 February-3 March 2017 (right). The red dots indicate pixels with the coldest tops ($\leq$ 190 K) that capture the deepest part of convection. The dashed circle indicates a range ring of 1000 km around the Maïdo Observatory (blue star).

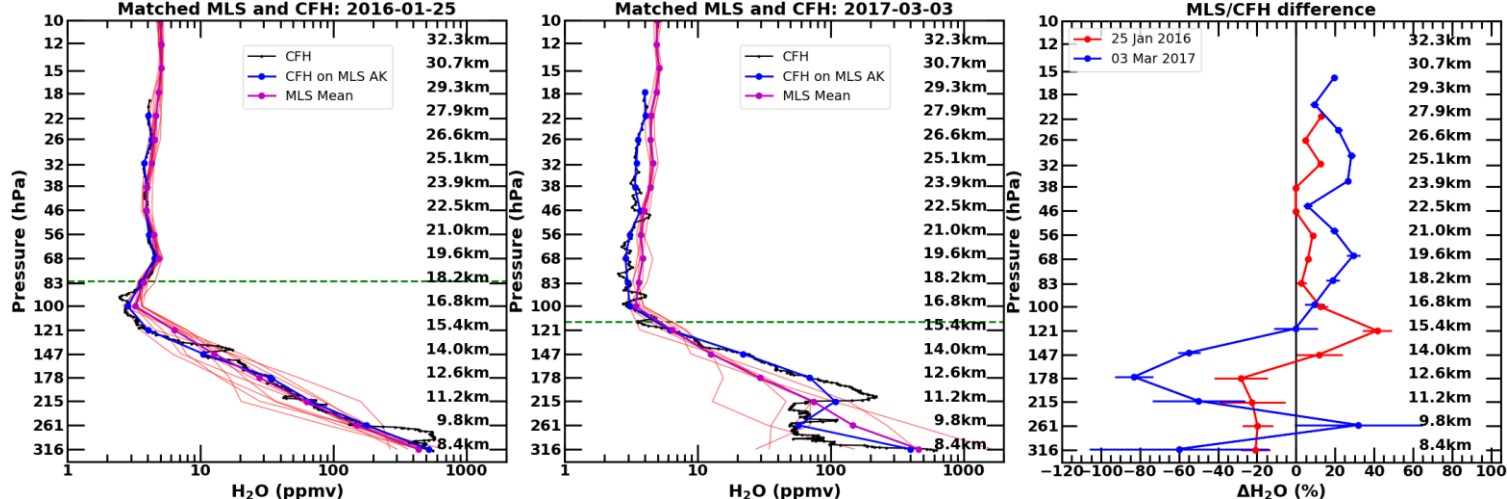

**Figure 9**: Left and middle panels: High-resolution (black line) and convolved (blue line) CFH water vapor profiles and closest-matched MLS profiles (thin red line) on 25 January 2016 (10 profiles) and 3 March 2017 (6 profiles). The mean MLS profile for each date corresponds to the thick magenta line. The location of the cold point tropopause is indicated by the dashed green line. Right panel: Mean percent difference between the convolved CFH water vapor profile and MLS coincident profiles on 25 January 2016 (red line) and 3 March 2017 (blue line). The horizontal bars indicate twice the standard error of the mean percent difference. Markers for each pressure level on 3 March 2017 are slightly offset in pressure for clarity. Corresponding altitude values for MLS pressure levels are also shown on each plot.

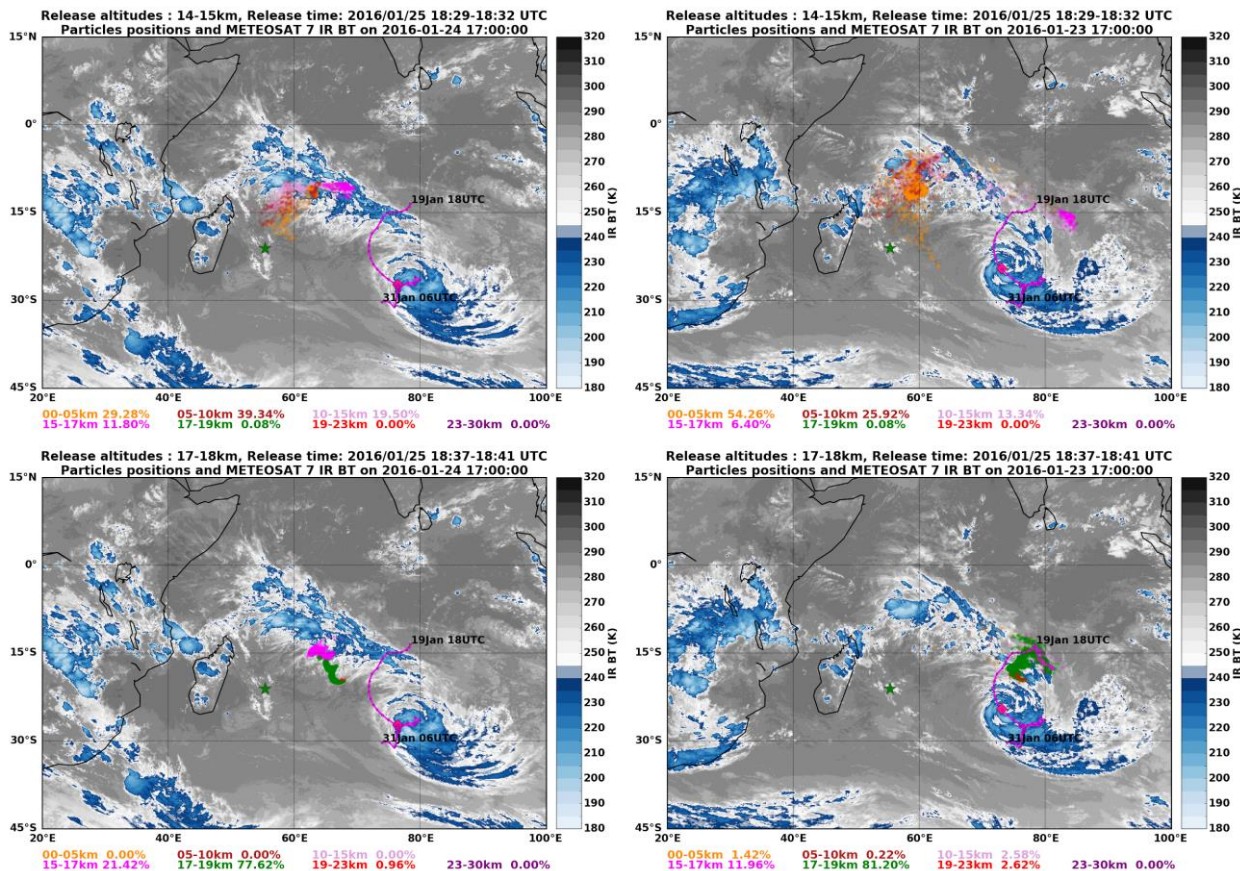

**Figure 10**: Backward trajectories calculated with the FLEXPART model for the CFH flight on 25 January 2016. On the upper panel backward trajectories were initialized for the 14-15 km layer (~147 hPa) on 25 January 2016. The particle positions one day before (on 24 January 2016 at 17 UTC, upper left) and two days before (on 23 January 2016 at 17 UTC, upper right) are shown with respect to the METEOSAT7 cloud distribution at those times. The altitude range of the particles (e.g. 0-5km) and the percent of particles in that altitude range are indicated according to a color code shown on the bottom of each panel. Bottom panel: same as upper panel but for backward trajectories initialized for the 17-18 km (83-100 hPa) layer on 25 January 2016.



**Figure 11**: Backward trajectories calculated with the FLEXPART model for the CFH flight on 3 March 2017. On the upper panel backward trajectories were initialized for the 14-15 km layer (~147 hPa) on 3 March 2017. The particle positions one day before (on 2 March 2017 at 17 UTC, upper right) and two days before (on 1 March 2017 at 17 UTC, upper left) are shown with respect to the METEOSAT7 cloud distribution at those times. The altitude range of the particles (e.g. 0-5km) and the percent of particles in that altitude range are indicated according to a color code shown on the bottom of each panel. Bottom panel: same as upper panel but for backward trajectories initialized for the 17-18km (83-100 hPa) layer on 3 March 2017.



**Figure 12:** Upper panel: Convolved CFH water vapor profiles (blue line), mean of closest-matched MLS profiles (magenta)
1005    and monthly mean climatological MLS water vapor profile for Réunion Island (black line, see text for definition of the MLS
climatological profile) on 25 January 2016 (upper left) and 3 March 2017 (upper right). The gray shaded area indicates the
interquartile range of the MLS climatology. The horizontal bars in black correspond to the one standard deviation range.
Bottom panel: Relative difference between the convolved CFH water vapor profile and the MLS climatological profile for
Réunion Island (blue line) and the mean of closest-matched MLS profiles and the MLS climatological profile (magenta line).
1010    Corresponding altitude values for MLS pressure levels are also shown on each plot.