# Peer review of "Effect of deep convection on the TTL composition over the Southwest"

_Atmospheric Chemistry and Physics, 2019_

## Referee Comment (RC1) · Anonymous Referee #1 · 15 Apr 2020

Review of:

**Effect of deep convection on the TTL composition over the Southwest Indian Ocean during austral summer.**

by Stephanie Evan, Jerome Brioude, Karen Rosenlof, Sean. M. Davis, Holger Vomel, Damien Heron, Francoise Posny, Jean-Marc Metzger, Valentin Duflot, Guillaume Payen, Helene Veremes, Philippe Keckhut, and Jean-Pierre Cammas

Anonymous Reviewer

**1 General Comments**

The authors present a study focussed on the origin of convective signatures on two balloon-borne water vapour profiles observed in the vicinity of tropical storms in January 2016 (TS Corentin) and March 2017 (TC Enawo). Additional ozone sondes, ground- and space-based Lidar data as well as Meteosat 7 IR images and AURA-MLS water vapour data have been utilized
5    and compared to the balloon data as well as model results from FLEXPART trajectory studies based on regular and nested high-resolution ECMWF analysis data. Therefore quite a suite of tools has been deployed in order to corroborate and interpret the measurements. Significant hydrations have been identified in the upper troposphere (UT) for both flights which based on the back trajectories could be traced to convective activity of the respective tropical storms. One dehydrated layer slightly below the cold point was identified. A slight moistening signal in the lower stratosphere (LS) has been found for the 2016 measurement
10   and could be related to horizontal transport from the SE-Indian ocean which was still humid in response to the strong ElNino. These major results are in-line with previously published results so no new or rare types of signatures are reported. Also, quite a number of modelling and experimental studies on this topic have been published over the years based on balloon-borne, airborne as well as satellite data. However, the availability of reliable and spatially well resolved data to discriminate different convective models and represented processes still seems not sufficient which raises the value of each new dataset being studied
15   and published. The study presented here focusses on the region of the South-West Indian Ocean which has not received a lot of attention with this respect.

As the role of moistening and drying effects of tropical storms on the UT, TTL, and LS are still under discussion this is a relevant and well suited topic for publication in ACP. The manuscript is well written, however the structure needs improvement. Data validation and interpretation are mixed up at places which makes the argumentation quite hard to follow. The number
20   of figures could also be reduced. As discussed in detail below, the data presented are not sufficient to conclusively support

several of the topics discussed in length in the manuscript. As a result I recommend to focus the discussion to the conclusively identified source-signature relations mentioned above and submit a revised and more focussed manuscript taking into account the comments below.

**2 Detailed Comments**

**l.100:** Apart from the CFH specifications the altitude dependent temperature measurement accuracy of the radio sonde must be given in order to judge the reliability of the relative humidity (RH) used later. Derivation and specifications of the measured vertical coordinate(s) should also be mentioned briefly.

**Structure of the Data/Results sections:** I recommend to move all validations (instrument intercomparisons) and derivations of climatological differences to one dedicated section where they are derived and presented without discussing them along with interpretational details. The argumentation would be much easier to follow when giving/deriving temperature differences, water vapour intercomparisons, RH, differences to monthly mean MLS $H_2O$ profiles first and then later discuss them all together with respect to the FLEXPART trajectory model results for the consecutive atmospheric layers and flights. Also I propose to discuss the possible CFH lower strat. dry bias on 3.March 2017 in this early section and just mention it later in the interpretation. In this last aspect it is also important to point out the reliability of the CFH results up into the UT supported by the good agreement with the lidar.

**CFH-MLS intercomparison:** The efforts to validate and intercompare the data employed for the study are very welcome! The few (6 and 10) MLS profiles selected employing the given criteria and presented in Fig.9 may, however, not be the best to compare to the CFH data since temporal and/or spatial vicinity alone doesn't really help. These single MLS profiles may very closely resemble the air masses probed by the CFHs on some altitudes but have very different origin for others. Therefore a low number of profiles will not be representative even when regarding the std.devs. I wonder if searching for matches for the measured air particles in the CFH profiles with previous or later MLS measurements in a given time window employing the Flexpart trajectories wouldn't do a much better job and provide at least a similar number of satellite measurements with less 'variability'. On a related issue: The stirring provided by the storms generates a quite inhomogeneous atmosphere up into the TTL (Cairo et al. (2008) and references therein). The spot measurement by CFH may by just hitting a fresh tropospheric filament yield humidities much higher than any MLS point ever could, and the other way around. Employing the averaging kernel of MLS on such an in-situ profile in a very inhomogeneous atmosphere will with low probability yield a good comparison even with a good spatial match. This applies also to Fig.12 and the discussion. I think the significance of any features can only be discussed within an at least $3 \times \sigma$ uncertainty in mixing ratio differences (or relative difference, Fig. 12). Therefore some of the smaller features presented may be somewhat over interpreted in the lengthy discussions that at the end do not yield a firm conclusion (wherever there are 'mays'. These parts should be severely shortened or even cut out. Also for the intercomparison and discussions the selection of MLS profiles is somewhat intransparent, e.g. for Fig. 12. This procedure should be detailed. For the

calculation of the monthly climatological average the proper way would be to exclude all MLS profiles which are probably affected by convection. Low UTLS ozone values could provide an indicator for those (e.g. Paulik and Birner, 2012).

**Flexpart simulations:** Why are the back trajectory studies for both flights limited to (or just only shown for?) the same two altitude layers? They are not particularly well chosen for either of the flights, e.g. for 25.Jan the 14-15km layer coincides with the crossover of $\Delta H_2O$ from moistening to drying in Fig.12. Why isn't a trajectory parcel initiated e.g. for each data point shown in Fig. 12. Then the percentage of trajectories originating from certain altitude regimes in the vicinity or away from the active storm influence regions could be shown. Figs. 10 and 11 present an interesting way of doing a visual analysis but this is extremely hard to relate to the observations. I think at least a proper adjustment of the analysis layers must be done/shown.

**l. 696ff:** Since the need for accurate and spatially well resolved data on tropical storm events is stressed later in the summary I would like to recommend adding ozone sondes and possibly backscatter sondes to the CFH payloads for future measurements as deployed in e.g. Li et al. (2017). Inclusion of BS sondes would certainly be a major additional cost and weight factor (reagrding that the instruments will be lost) but give very valuable information on the condensed water phase which here can only be approximated from the CALIPSO tracks. However, also regarding the major efforts in flight planning, ozone sondes would offer a very cost effective option yielding really parallel profiles which would help to better identify air mass origins. Here independent ozone profiles from sondes launched on different dates are utilized which is generally a good add-on but can't replace parallel ozone measurements for the interpretation of observed signatures (in absence of other tracer information). E.g. RH shows a very structured profile which is obviously driven by the temperature profile for 25.Jan.2016, however the RH structure on 03.Mar.2017 is not as obvious from the observed temperature (Fig.7). Here an online ozone profile would be extremely valuable to discriminate on air mass origin signatures. The 4.Feb. ozone profile nicely shows the remaining signatures of Corentin in contrast to 18.Jan. but is useless to discuss detailed dynamical features of the $H_2O$ profile.

**3   Figures**

**Fig.1:** I think it would be worthwhile to also show the actual flight tracks of the CFH sondes.

**Fig.2:** This figure is trivial and could be dropped. The explanation given in the text is sufficient. The nested region could still be given by a rectangle in Fig.1.

**Figs.3+4:** The climatological fields in the top panels of Figures 3 and 4 are not really needed as the respective climatologies for the actual profiles are given and discussed later on. The bottom panels are useful giving insight into the monthly mean situations at UT and TTL levels and could be combined into one figure then.

**Fig.5:** It might be useful to indicate (shade) the most important signatures/layers and name them to help in the discussion (possibly also in other related figures...). Caption: Is the std. dev. shown 1sigma? Please specify.

**Fig 6.:** It would be sufficient to show the CALIPSO track only up to 22-23S (this would enable to blow up the figure in the vertical). Please align the vertical axes on left and right panels, possibly also show the CPT as a line on the left panels!

**Fig.7:** The upper panels do not give essential additional information over the text and the derived temperature differences to the climatology could be integrated into the left panels of Fig.6.

**Figs. 10.+11.:** See general comment above ...

**4  Typos etc.**

**l.95:** ...Frostpoint...

**l.242:** ... at 215hPa in for ..., remove "in"

**l.282:** The lidar smoothes ...

**l.303:** ...north of Madagascar ...

**l.502:** ... due to the 0.5x0.5 ...

**l.517:** "Figure 10b", harmonize with the caption (either "upper right, etc." or a, ,b, c, d)

**l.525:** On 23 January ...

**l.626:** ... on the water vapour mixing ratio at 100hPa ... (slang)

**l.634:** ... and the ... is less important ... (meaning unclear)

**l.650:** ... impact of the ...

**References**

Cairo, F., Buontempo, C., MacKenzie, A. R., Schiller, C., Volk, C. M., Adriani, A., Mitev, V., Matthey, R., Di Donfrancesco, G., Oulanovsky, A., Ravegnani, F., Yushkov, V., Snels, M., Cagnazzo, C., and Stefanutti, L.: Morphology of the tropopause layer and lower stratosphere above a tropical cyclone: a case study on cyclone Davina (1999), Atmos. Chem. Phys., 8, 3411–3426, https://doi.org/10.5194/acp-8-3411-2008, 2008.

Li, D., Vogel, B., Bian, J., Mueller, R., Pan, L. L., Gunther, G., Bai, Z., Li, Q., Zhang, J., Fan, Q., and Voemel, H.: Impact of typhoons on the composition of the upper troposphere within the Asian summer monsoon anticyclone: the SWOP campaign in Lhasa 2013, Atmos. Chem. Phys., 17, 4657–4672, https://doi.org/10.5194/acp-17-4657-2017, 2017.

---

## Referee Comment (RC2) · Anonymous Referee #2 · 24 Apr 2020

**General comments:**

The paper analyses the impact of two case studies of tropical cyclones (Corentin and Enawo) over the Indian Ocean on the TTL composition, with a particular focus on the water vapor consequent anomalies. In both cases, the authors identified positive anomalies of water vapor in the upper troposphere that could be traced back to the convective activities linked with the tropical cyclones. In the Corentin case the balloon launch revealed also a dry layer around 100hPa and a wet anomaly at 68 hPa (linked to transport of wet El Nino influenced-air from the South East Indian Ocean) while the second balloon launch did not reveal any significant perturbation near or above the tropopause from the tropical cyclone Enawo. The paper presents a detailed description of the hydration/dehydration impact of the two events making use a variety of observations and trajectory studies and gives a comprehensive analysis of the possible contributing processes. I agree on the publication of the paper in ACP with some revisions required.

One main concern is on the structure of the paper that, at the present state, makes very difficult to follow the logic of the work. The paper in fact is made of several long and detailed sections that sometimes are missing a clear statement on which is the main information to retain. The sections are therefore hard to connect and this is also made more difficult by their organization itself. I would advise to put in an uninterrupted sequence all the sections regarding the profile measurements for the two events, including the FLEXPART study (that is indeed supporting the analysis and is instead put toward the end of the paper). The monthly and climatological water vapor distributions and the CFH and MLS comparison can be on separated sections or moved elsewhere in a way to not interrupt the logic of the two events analysis. Some sections are very long as well, like the 4.4. that puts together the RHice analysis, the temperature anomalies and the distribution of deep convective clouds, it may be worth to separate them in subsections.

The abstract is lacking a highlight on the scientific impact from the main findings. What can we conclude on the TTL hydration by deep convection from the two events analysis?

In addition, I found that the figures are often not referenced when due, and that implies an extra effort for the reader to figure out which panels or which figure the statements are referring to. I would advise to check in the data description paragraphs and add a precise reference to the plot (and panel) that is being described.

**Specific comments:**

Line 54 page 2: can you add few references?

Line 107 page 4: Do you identify the convection from the Lagrangian forecasting tool in the forecast mode in the same way as explained later for the analysis? How do you use the meteosat-7 information (that are in the "past") for the forecast of the storm position?

Figure1 and/or line 202 page 7: Can you give a brief definition of what you mean by "best track"?

Line 204 page 8: How do you get the pressure at the TS center?

Line 242 page 8: You should rephrase here. Looking to the upper left panel of Figure 3 and the lower left one (January 2016) the mixing ratios do not really seem in agreement, with differences in the whole longitude band between 50 and 150 E.

Line 257 page 9: Add reference to the panels you are comparing. If you are talking about the two upper panels this difference of 0.43 ppmv is not visible (as instead between the two lower panels). How do you compute this difference?

Line 263 page 10: Why are you specifically mentioning here just December 2015? Is it because it corresponds to the maximum anomalies in water vapor?

Lines 282-284 page 10: This statement is not very convincing. Can we really say that the 9-14 km layer is a moist one when the observations show only a small peak around 10 km?

Line 320 page 12: I would really help to have a plot of the brightness temperature with the CALIOP track and the wind direction / geopotential to show the mean circulation pattern, same for the March case.

Lines 505-508 page 18: The sentence, as is presented now, is not really giving an indication on the capability of the trajectory method in the convective origin study. Do you have some references indicating the quality of ECMWF 0.15x0.15 analysis in resolving vertical velocities for tropical cyclones? Also, I think this paragraph is better fitting in the method presentation of section 3.

Lines 577 – 579 page 20: This sentence is confusing. Do you mean the difference averaged between 316 and 261 hPa is -20% for both days and for both CFH and MLS mean? That does not seem correct.

Line 650 page 22: "The QBO easterlies can be observed at 70 hPa"…from where?

**Technical corrections:**

Line 15 page 1: It's worth to specify also in the abstract what CFH stands for.

Line 57 page 2: the upper 700m of what, the sea surface?

Line 58 page 2 : .. that convection deeper than 15 km…

Line 156 page 6: Do you mean: "..cross section. More details are given in the CALIOP Algorithm …"?

Line 229 page 8: Latitude / longitude grid (5°x20° resolution)

Line 230 page 8: at 215 hPa (figure 3) and 100 hPa (figure 4) for January 2016 (lower left panel) as March 2017 (lower right panel)

Line 269 Page 10: The red and purple lines…

Lines 340-341 page 12: This is one example of a needed figure reference. Does it refer to figure 7?

Line 354 page 12: reference to bottom left panel of figure 7?

Line 650 page 22: …the impact OF the 2016 strong ..

Figure1: The green star on the grey background is not very easy to spot! Similarly for the dates label that are black and with a small font.

Figure 5: Please, add a legend for the lines, it will ease the reading of the plot.

Figure 10: The so called "brown dots" are difficult to distinguish from the red ones. The light pink is instead not very visible.

Figure 11: The panels notation must be homogeneous. I would suggest indeed to reference the panels with letters, as done here, since it makes the reading more fluid. Same thing with the label "-1 day" "-2 days" that are missing on figure 10.

---

## Author Comment (AC1) · 11 Jun 2020

We thank the reviewer for her/his helpful comments on the manuscript. In the following, reviewer's comments are in italics, authors' responses are in normal font. Numerous figures are shown in our response to illustrate our points but are not included in the revised manuscript.

**1. General Comments.**

*As the role of moistening and drying effects of tropical storms on the UT, TTL, and LS are still under discussion this is a relevant and well suited topic for publication in ACP.*

We thank the reviewer for the positive assessment.

*The manuscript is well written, however the structure needs improvement. Data validation and interpretation are mixed up at places which makes the argumentation quite hard to follow. The number of figures could also be reduced. As discussed in detail below, the data presented are not sufficient to conclusively support several of the topics discussed in length in the manuscript. As a result I recommend to focus the discussion to the conclusively identified source-signature relations mentioned above and submit a revised and more focussed manuscript taking into account the comments below.*

We agree that the structure of the paper needed improvement and that it could be shortened/more focussed. The number of figures has been reduced from 12 to 9 in the revised manuscript. We have changed the overall structure of the manuscript to better separate the description of observations/Lagrangian simulations and the discussion of the general results. We have restricted our analysis on essential results and shortened the data validation/interpretation as suggested by the referees.

**2. Detailed Comments.**

*l.100: Apart from the CFH specifications the altitude dependent temperature measurement accuracy of the radiosonde must be given in order to judge the reliability of the relative humidity (RH) used later. Derivation and specifications of the measured vertical coordinate(s) should also be mentioned briefly.*

The iMET-1-RSB has a temperature measurement uncertainty of 0.3°C, or 5% in RH, with an altitude independent bias of $0.5 \pm 0.2$ °C (Hurst et al., 2011). As for vertical coordinate, we use the geopotential height calculated from the iMet-1-RSB measurements of pressure, temperature and RH. Hurst et al. (2011) reported altitude-dependent differences of -0.1 to -0.2 km above 20 km between the geopotential altitudes derived from the Vaisala RS92 and Intermet iMet-1-RSB sondes. This description has been added in section 2.1 (Balloon data) of the revised manuscript.

***Structure of the Data/Results sections:*** *I recommend to move all validations (instrument intercomparisons) and derivations of climatological differences to one dedicated section where they are derived and presented without discussing them along with interpretational details. The argumentation would be much easier to follow when giving/deriving temperature differences, water vapour intercomparisons, RH, differences to monthly mean MLS H2O profiles first and then later discuss them all together with respect to the FLEXPART trajectory model results for the consecutive atmospheric layers and flights. Also I propose to discuss the possible CFH lower strat. dry bias on 3 March 2017 in this early section and just mention it later in the interpretation. In this last aspect it is also important to point out the reliability of the CFH results up into the UT supported by the good agreement with the lidar.*

As suggested by the reviewer, we have reorganized the manuscript. The section that describes the FLEXPART model is now section 2.4. In section 3, TS Corentin and TC Enawo are described (Figure 1 of the revised manuscript), the mean convective cloud cover is presented (Figure 2) and the MLS water vapor mixing ratio gridded in the SWOOSH data set at 215&100hPa averaged over January 2016 and March 2017 (Figure 3).

In section 4, the water vapor and ozone profiles are described (section 4.1, Figure 4), the relative humidity and temperature profiles (section 4.2, Figure 5) and the Lagrangian analysis of air mass origin with FLEXPART (section 4.3, Figures 6 and 7). The MLS and CFH comparison is presented in section 5.1 (Figure 8) and temperature anomalies derived from the NDACC/SHADOZ radiosonde dataset are discussed in section 5.2 (Figure 5). Finally, the water vapor anomalies, derived from the MLS climatology are discussed in section 5.3 (Figure 9). The dry bias found in the lower stratosphere has been reduced by restricting the MLS profiles to those directly impacted by TC Enawo in the revised manuscript.

*CFH-MLS intercomparison: The efforts to validate and intercompare the data employed for the study are very welcome! The few (6 and 10) MLS profiles selected employing the given criteria and presented in Fig.9 may, however, not be the best to compare to the CFH data since temporal and/or spatial vicinity alone doesn't really help. These single MLS profiles may very closely resemble the air masses probed by the CFHs on some altitudes but have very different origin for others. Therefore a low number of profiles will not be representative even when regarding the std.devs. I wonder if searching for matches for the measured air particles in the CFH profiles with previous or later MLS measurements in a given time window employing the Flexpart trajectories wouldn't do a much better job and provide at least a similar number of satellite measurements with less 'variability'.*

As you suggested, we have used FLEXPART backward trajectories to better identify MLS coincident profiles for the comparison with the CFH measurements. We agree that some MLS profiles may very closely resemble the air masses probed by the CFHs on some altitudes but have very different origins for others. The map below shows the location and time of the 10 and 6 MLS profiles initially used for comparison with the CFH data on 25 January 2016 and 3 March 2017 respectively. These profiles correspond to match criteria of ±18h, ±500 km North-South distance (around ±5° latitude), ±1000 km East-West distance (around ±10° longitude).

[Figure]

Figure 1: MLS coincident profiles on 25 January 2016 (left) and 3 March 2017 (right). The green square indicates the location of the Maïdo Observatory on Réunion Island.

We initialized FLEXPART backward trajectories for each of the locations shown on the map above and for each of the 19 MLS pressure levels between 316 and 10hPa. FLEXPART backward trajectories were also run from the location of the CFH measurements and similar pressure levels as MLS. This was done to ensure that we select MLS profiles that have the same origin as the air masses sampled by the CFH on 25 January 2016 and 3 March 2017.

FLEXPART can provide gridded output of the residence time (in seconds) as a function of latitude/longitude/altitude. We ran backward simulations over 1 week and the residence time field was reported on a regular latitude/longitude grid with 0.25°x0.25° resolution, 1 km vertical spacing from the surface to 25 km and every 3 hours. The gridded residence time can be summed in time and altitude to indicate the total time an air parcel resides at a given grid point for a given period/altitude range. For example, the residence summed over the altitude range 0-5 km will indicate how much time an air parcel stayed in the low troposphere. Figure 2, below, shows the 0-5 km residence time of 50,000 trajectories released at 178 hPa from 6 different locations which are the locations of the CFH on 25 January 2016 (upper left)/3 March 2017 (upper right), MLS profile on 25 Jan at 21:39UTC (so-called MLS03, middle left), MLS profile on 04 Mar at 10:21UTC (MLS 04, middle right), MLS profile on 26 Jan at 09:49UTC (MLS 07, bottom left) and MLS profile on 04 Mar at 10:20UTC (MLS03, bottom right). The red contours indicate residence time summed over 0-5km and 2 days before the launch. Thus, the pattern on each panel corresponds to the principal region of origin in the low troposphere for air parcels initially released at 178 UTC. We produce similar maps for all MLS locations (shown on Figure 1 above)/19 pressure levels and the CFH launch location/pressure levels. The residence time was summed over altitude layers 00-05km (low troposphere), 05-10km (middle troposphere) and 10-15km (upper troposphere) and a period of 2 days before the launch. Thus, a total of 11x19x3=627 and 7x19x3=399 maps were

produced for 25 January 2016 and 3 March 2017 respectively but to illustrate our comparison we only show 6 maps of residence time on Figure 2. For each CFH launch, we then select coincident MLS profiles that have a common origin in the low troposphere as we are interested in convectively influenced profiles. For example, Figure 2 shows that MLS profile #3 (MLS03, 25/01 21:39UTC, middle left ) at 178hPa has the same lower tropospheric origin as the 25 Jan 2016 CFH (upper left) while MLS profile #7 (MLS07, 26/01 09:49UTC, bottom left) has an origin further east of 60°E. For the CFH on 3 Mar 2017, MLS profile #4 has the same origin north of 15°S while MLS profile #3 comes from the south of 20°S.

By comparing several maps of MLS/CFH residence time, we further restricted our selection of coincident profiles: 5 for 25 January 2016 (MLS02, 03, 04, 05, 06 in blue on Figure 1) and 3 for 3 March 2017 (MLS04, 05, 06 in blue on Figure 1).

[Figure]

Figure 2: Total residence time in the low troposphere (0-5km) and over 2 days for CFH/MLS locations on 25/26 January 2016 (left column) and 3/4 March 2017 (right column). The residence time was computed using bactrajectories initialized at 178hPa. The green squares indicate the location of the Maïdo Observatory and the blue squares indicate the position of the MLS/CFH profiles.

*On a related issue: The stirring provided by the storms generates a quite inhomogeneous atmosphere up into the TTL (Cairo et al. (2008) and references therein). The spot measurement by CFH may by just hitting a fresh tropospheric filament yield humidities much higher than any MLS point ever could, and the other way around. Employing the averaging kernel of MLS on such an in-situ profile in a very inhomogeneous atmosphere will with low probability yield a good comparison even with a good spatial match. This applies also to Fig.12 and the discussion. I think the significance of any features can only be discussed within an at least 3×σ uncertainty in mixing ratio differences (or relative difference, Fig. 12). Therefore some of the smaller features presented may be somewhat over interpreted in the lengthy discussions that at the end do not yield a firm conclusion (wherever there are 'mays'. These parts should be severely shortened or even cut out. Also for the intercomparison and discussions the selection of MLS profiles is somewhat intransparent, e.g. for Fig. 12. This procedure should be detailed.*

Thank you for pointing out the Cairo et al. (2008) study. We have added this reference to the revised manuscript. As you said, differences in CFH and MLS coincident profiles (using the match criteria+FLEXPART trajectories) can still arise because of the mixing in the troposphere induced by the tropical cyclone. We have clarified this point in the description of Figure 8 of the revised manuscript (formerly Figure 9). Please see section 5.1 (CFH and MLS comparison). We have also significantly shortened the description of the differences, please see section 5. 1..

*For the calculation of the monthly climatological average the proper way would be to exclude all MLS profiles which are probably affected by convection. Low UTLS ozone values could provide an indicator for those (e.g. Paulik and Birner, 2012).*

[Figure]

Figure 3: Upper troposphere (261 to 147 hPa) MLS ozone distribution for January 2005-2017 (left) and March 2005-2017 (right). The red numbers correspond to the mean and 25th/75th percentiles of the distribution.

Figure 3, above, shows the upper tropospheric distribution of MLS ozone for 10 January-9 February 2005-2017 and 16 February-18 March 2005-2017. We use the 25th&75th percentiles of the upper tropospheric ozone distribution to classify MLS ozone profiles as low-ozone (mean 261-147hPa Ozone < 25th percentile of the entire distribution), close to average ozone (25th percentile of the entire distribution < mean 261-147hPa Ozone < 75th percentile of the entire distribution) and high-ozone (mean 261-147hPa Ozone > 75th percentile of the entire distribution). As you suggested and as shown for example by Paulik and Birner (2012), low upper-tropospheric ozone is a sign of a convective influence. We have further classified the associated MLS water vapor profiles in 3 categories corresponding to the low, close to average and high upper-tropospheric ozone. A monthly climatological water vapor profile is computed for each category. By keeping the MLS water vapor profiles with ozone close to the average climatological value, we make sure that we select profiles that most likely weren't influenced by convection. The result is shown on Figure 4 below.

[Figure]

Figure 4: Monthly mean climatological MLS water vapor profile for Réunion Island for January (left) and March (right). On each plot, the 4 water vapor profiles correspond to low upper-tropospheric ozone (magenta), close to average upper-tropospheric ozone (green), high upper-tropospheric ozone (blue) and the climatological profile using all profiles (black).

The non-convective and monthly climatological MLS water vapor profile (using all profiles) look very similar (green and black curves on Figure 4, note that it is quite difficult to distinguish the two curves). Thus, the climatological MLS water profile using all profiles is used for comparison with the water vapor measurements on 25 January 2016 and 3 March 2017. This was added to the revised manuscript, section 5.3 (Water vapor anomaly).

***Flexpart simulations:*** *Why are the back trajectory studies for both flights limited to (or just only shown for?) the same two altitude layers? They are not particularly well chosen for either of the flights, e.g. for 25.Jan the 14-15km layer coincides with the crossover of ΔH2O from moistening to drying in Fig.12. Why isn't a trajectory parcel initiated e.g. for each data point shown in Fig. 12. Then the percentage of trajectories originating from certain altitude regimes in the vicinity or away from the active storm influence regions could be shown. Figs. 10 and 11 present an interesting way of doing a visual analysis but this is extremely hard to relate to the observations. I think at least a proper adjustment of the analysis layers must be done/shown.*

Figures 10&11 showing the FLEXPART back trajectories have been redone. They are now Figures 6&7 in the revised manuscript. We ran FLEXPART trajectories for all MLS pressure levels shown on Figures 8&9 of the revised manuscript. Trajectories (50,000) are initialized at the location of the Maïdo Observatory for each MLS pressure level between 316 and 10 hPa and run backward in time for two weeks. The positions of the back trajectories are outputted every 3 hours. In the revised manuscript, Figures 6&7 show the positions of trajectories 2 and 3 days before the CFH launch date, i.e. 25 January 2016 for Figure 6 and 3 March 2017 for Figure 7. We now show the positions of trajectories released at pressure levels 178 hPa (~12.6km, top panels of Figures 6&7) and 100 hPa (~16.8 km, bottom panels of Figures 6&7). The pressure level 178 hPa corresponds to layers L1 and L4 on Figures 4, 8&9 of the revised manuscript. These layers correspond to moistening/low ozone. The pressure level 100 hPa corresponds to layers L2 and L5 on Figures 4, 8&9 (25 January 2016 and 3 March 2017 respectively). They correspond to regions of drying. We did also produce maps of infrared brightness temperature/trajectories' locations for layer L3 (68 hPa/19.6) on 25 January 2016 but they are not included in the revised manuscript in order to keep the number of figures to a minimum.

As you suggested, we now include an estimate of convective influence for each of the 19 pressure levels between 316 hPa and 10 hPa. Each of the 50,000 trajectories released at a pressure level is examined for coincidence with deep convection. The infrared brightness temperature (IR BT) data from METEOSAT 7 are used as a proxy of deep convection and are interpolated along the trajectories. Hence, the change of IR BT can be studied along the trajectories. Air parcels are tagged as convectively influenced, when the IR BT observed by METEOSAT 7 along their back trajectories falls below 230 K (we also used thresholds of 210 and 230 K but the results are pretty similar). A second check is that the altitude along the back trajectories falls below 5 km, indicating a lower tropospheric origin.

Figure 5 shows an example of altitude/IR BT evolution as a function of time (backward) along a trajectory released at 178 hPa on 25 January 2016. The initial position in altitude is ~12.6 km and IR BT ~240K. After 25h of backtrajectory, both altitude and IR BT decrease. The air parcel experienced a rapid change in altitude (from ~12 to 1km) as the IR BT falls below 230K. Thus this trajectory can be tagged as convectively influenced. By applying the 230 K and 5 km criteria to all trajectories at a given pressure level, we can identify which trajectory is convectively influenced. The convective fraction of a layer is then defined as the ratio of the number of convectively influenced trajectories to the total number of trajectories (i.e. 50,000). It is shown on Figure 9 of the revised manuscript for both CFH flights (red line on Figure 9). The definition of convective fraction shows little sensitivity to the IR BT threshold used to identify deep convection (210, 220 or 230 K) as shown on Figure 6 below.

[Figure]

Figure 5: Example of altitude/Infrared brightness temperature along a backward trajectory released at 178 hPa on 25 January 2016 18UTC. Time (in hours) is relative to the beginning of the trajectory and the symbols indicate values every 3 hours.

[Figure]

Figure 6: Sensitivity of the convective fraction to the IR BT threshold used to define deep convection: 210K (left), 220K (middle) and 230K (right).

*l. 696ff: Since the need for accurate and spatially well resolved data on tropical storm events is stressed later in the summary I would like to recommend adding ozone sondes and possibly backscatter sondes to the CFH payloads for future measurements as deployed in e.g. Li et al. (2017). Inclusion of BS sondes would certainly be a major additional cost and weight factor (regarding that the instruments will be lost) but give very valuable information on the condensed water phase which here can only be approximated from the CALIPSO tracks. However, also regarding the major efforts in flight planning, ozone sondes would offer a very cost effective option yielding really parallel profiles which would help to better identify air mass origins. Here independent ozone profiles from sondes launched on different dates are utilized which is generally a good add-on but can't replace parallel ozone measurements for the interpretation of observed signatures (in absence of other tracer information). E.g. RH shows a very structured profile which is obviously driven by the temperature profile for 25.Jan.2016, however the RH structure on 03.Mar.2017 is not as obvious from the observed temperature (Fig.7). Here an online ozone profile would be extremely valuable to discriminate on air mass origin signatures. The 4.Feb. ozone profile nicely shows the remaining signatures of Corentin in contrast to 18.Jan. but is useless to discuss detailed dynamical features of the H2O profile.*

We agree that adding an ozonesonde and a backscatter sonde to the CFH payload is important to better describe the effect of cirrus clouds/deep convection on the TTL over the SWIO. Initial CFH payloads did not include an ECC ozonesonde and backscatter sonde due to the cost. However, there is an ongoing project funded by the French National Research Agency (ANR) in October 2017 to study the effects of convection and cirrus clouds on the Tropical Tropopause Layer over the Indian Ocean (the CONCIRTO project:
https://anr.fr/en/funded-projects-and-impact/funded-projects/project/funded/project/b2d9d3668f92a3b9fbbf7866072501ef-3a188e016e/?tx_anrprojects_funded%5Bcontroller%5D=Funded&cHash=975f599a3b6d31bf21638b8272c285db)
This project aims to further our understanding of deep convection and cirrus clouds and how they affect the TTL over the Indian Ocean. As you know, balloon sondes, a cost-effective mean to study TTL processes compared to high-altitude aircraft that can reach the TTL, reveal fine-scale features that are below the vertical resolution of satellite sounding systems. The CONCIRTO project has funded coincidental high-resolution balloon in situ measurements of water vapor (CFH), ozone (ECC ozonesonde), aerosol (COBALD: Compact Optical Backscatter Aerosol Detector, e.g. Brabec et al., 2012; Brunamonti et al., 2018)/ice particles with a Doppler polarimetric cloud radar observations at the Maïdo Observatory. A total of 10 soundings with CFH/COBALD/ECC ozonesondes were performed in Austral Summers 2018/2019/2020 to target convective events (e.g. tropical cyclones) and cirrus clouds. The results of the CONCIRTO project will be the subject of a subsequent study.

**3. Figures.**
*Fig.1: I think it would be worthwhile to also show the actual flight tracks of the CFH sondes.*

We modified Figure 1 following the recommendation of Referee #2. It now shows in addition to the TS Corentin and TC Enawo's best tracks/METEOSAT7 Infrared Brightness temperature, the CALIPSO's tracks on 25 January 2016 and 3 March 2017 (yellow lines) and the ECMWF winds&geopotential heights at 150 hPa. We did try to include the flight tracks of the CFH sondes but because of the scale of the plot, they appeared as a point on Figure 1. Adding an extra figure with a zoom on Réunion Island to only show the CFH flight tracks would increase the number of figures in the manuscript.

*Fig.2: This figure is trivial and could be dropped. The explanation given in the text is sufficient. The nested region could still be given by a rectangle in Fig.1.*

The figure has been removed in the revised manuscript, instead the nested region is defined in the text.

*Figs.3+4: The climatological fields in the top panels of Figures 3 and 4 are not really needed as the respective climatologies for the actual profiles are given and discussed later on. The bottom panels are useful giving insight into the monthly mean situations at UT and TTL levels and could be combined into one figure then.*

Figures 3&4 have been combined to produce Figure Figure 3 in the revised manuscript.

*Fig.5: It might be useful to indicate (shade) the most important signatures/layers and name them to help in the discussion (possibly also in other related figures...). Caption: Is the std. dev. shown 1sigma? Please specify.*

Important layers have been shaded and named on Figures 4/8&9.

*Fig 6.: It would be sufficient to show the CALIPSO track only up to 22-23S (this would enable to blow up the figure in the vertical). Please align the vertical axes on left and right panels, possibly also show the CPT as a line on the left panels!*

Figures 6&7 have been modified accordingly to produce Figure 5 in the revised manuscript.

*Fig.7: The upper panels do not give essential additional information over the text and the derived temperature differences to the climatology could be integrated into the left panels of Fig.6.*
The figure has been combined with Figure 6 to produce Figure 5 in the revised manuscript

*Figs. 10.+11.: See general comment above ...*
They are now Figures 6&7 in the revised manuscript, please see our response to the comment "FLEXPART simulations" above.

**4. Typos**
All typos have been corrected.

**References:**
Brabec, M., Wienhold, F. G., Luo, B. P., Vömel, H., Immler, F., Steiner, P., Hausammann, E., Weers, U., and Peter, T.: Particle backscatter and relative humidity measured across cirrus clouds and comparison with microphysical cirrus modelling, Atmos. Chem. Phys., 12, 9135–9148, https://doi.org/10.5194/acp-12-9135-2012, 2012.

Brunamonti, S., Jorge, T., Oelsner, P., Hanumanthu, S., Singh, B. B., Kumar, K. R., Sonbawne, S., Meier, S., Singh, D., Wienhold, F. G., Luo, B. P., Boettcher, M., Poltera, Y., Jauhiainen, H., Kayastha, R., Karmacharya, J., Dirksen, R., Naja, M., Rex, M., Fadnavis, S., and Peter, T.: Balloon-borne measurements of temperature, water vapor, ozone and aerosol backscatter on the southern slopes of the Himalayas during StratoClim 2016–2017, Atmos. Chem. Phys., 18, 15937–15957, https://doi.org/10.5194/acp-18-15937-2018, 2018.

---

## Author Comment (AC2) · 11 Jun 2020

Reply to Anonymous Referee 2

We thank the reviewer for her/his helpful comments on the manuscript. In the following, reviewer's comments are in italics, authors' responses are in normal font.

**General comments:**

*The paper analyses the impact of two case studies of tropical cyclones (Corentin and Enawo) over the Indian Ocean on the TTL composition, with a particular focus on the water vapor consequent anomalies. In both cases, the authors identified positive anomalies of water vapor in the upper troposphere that could be traced back to the convective activities linked with the tropical cyclones. In the Corentin case the balloon launch revealed also a dry layer around 100hPa and a wet anomaly at 68 hPa (linked to transport of wet El Nino influenced-air from the South East Indian Ocean) while the second balloon launch did not reveal any significant perturbation near or above the tropopause from the tropical cyclone Enawo. The paper presents a detailed description of the hydration/dehydration impact of the two events making use a variety of observations and trajectory studies and gives a comprehensive analysis of the possible contributing processes. I agree on the publication of the paper in ACP with some revisions required.*

We thank the reviewer for the positive assessment.

*One main concern is on the structure of the paper that, at the present state, makes it very difficult to follow the logic of the work. The paper in fact is made of several long and detailed sections that sometimes are missing a clear statement on which is the main information to retain. The sections are therefore hard to connect and this is also made more difficult by their organization itself. I would advise to put in an uninterrupted sequence all the sections regarding the profile measurements for the two events, including the FLEXPART study (that is indeed supporting the analysis and is instead put toward the end of the paper). The monthly and climatological water vapor distributions and the CFH and MLS comparison can be on separated sections or moved elsewhere in a way to not interrupt the logic of the two events analysis. Some sections are very long as well, like the 4.4. that puts together the RHice analysis, the temperature anomalies and the distribution of deep convective clouds; it may be worth to separate them in subsections.*

In response to your comment and reviewer #1's, we have modified the order of the sections in the revised manuscript, and the length of the sections have been reduced. The section that describes the FLEXPART model is now section 2.4.
In section 3, TS Corentin and TC Enawo are described (Figure 1 of the revised manuscript), the mean convective cloud cover is presented (Figure 2) and the MLS water vapor mixing ratio gridded in the SWOOSH data set at 215&100hPa averaged over January 2016 and March 2017 (Figure 3).
In section 4, the water vapor and ozone profiles are described (section 4.1, Figure 4), the relative humidity and temperature profiles (section 4.2, Figure 5) and the Lagrangian analysis of air mass origin with FLEXPART (section 4.3, Figures 6 and 7). The MLS and CFH comparison is presented in section 5.1 (Figure 8) and temperature anomalies derived from the NDACC/SHADOZ radiosonde dataset are discussed in section 5.2 (Figure 5). Finally, the water vapor anomalies, derived from the MLS climatology are discussed in section 5.3 (Figure 9).

*The abstract is lacking a highlight on the scientific impact from the main findings. What can we conclude on the TTL hydration by deep convection from the two events analysis?*

We added a comment at the end that explains that the paper demonstrates the need for accurate balloon-borne measurements of water vapor/ozone/aerosols in regions where TTL in-situ observations are sparse.

*In addition, I found that the figures are often not referenced when due, and that implies an extra effort for the reader to figure out which panels or which figure the statements are referring to. I would advise to check in the data description paragraphs and add a precise reference to the plot (and panel) that is being described.*

The figures are now properly referenced.

**Specific comments:**

*Line 54 page 2: can you add a few references?*

We have added three references: Toon et al., 2010; Jensen et al., 2017; Brunamonti et al., 2018.

*Line 107 page 4: Do you identify the convection from the Lagrangian forecasting tool in the forecast mode in the same way as explained later for the analysis? How do you use the meteosat-7 information (that are in the "past") for the forecast of the storm position?*

We have added the following comments: During austral summer, balloon launch planning is optimized using a Lagrangian forecasting tool. 5-day backward Lagrangian trajectories initialized from the location of the Maïdo Observatory at different altitudes (9.5, 12.5, 15.5 and 18 km) are run twice-daily and superimposed on current geostationary infrared satellite images to identify on-going convection over the SWIO (http://geosur.univ-reunion.fr/foot).

*Figure1 and/or line 202 page 7: Can you give a brief definition of what you mean by "best track"?*

The best track represents the best guest of the location of the tropical cyclone center every 6 hours. This comment was added in the revised manuscript.

*Line 204 page 8: How do you get the pressure at the TS center?*

The pressure at the tropical cyclone's center was provided in the back track data provided by Météo-France for the SWIO. Basically, the pressure at the storm center is derived from an empirical relationship between maximum surface wind speed and pressure. The surface wind speed is estimated from satellite scatterometer data.

*Line 242 page 8: You should rephrase here. Looking to the upper left panel of Figure 3 and the lower left one (January 2016) the mixing ratios do not really seem in agreement, with differences in the whole longitude band between 50 and 150 E.*

In the corrected manuscript, those figures are no longer present.

*Line 257 page 9: Add reference to the panels you are comparing. If you are talking about the two upper panels this difference of 0.43 ppmv is not visible (as instead between the two lower panels). How do you compute this difference?*

In the corrected manuscript, those figures are no longer present. The text has been modified accordingly.

*Line 263 page 10: Why are you specifically mentioning here just December 2015? Is it because it corresponds to the maximum anomalies in water vapor?*

Yes, December 2015 has the maximum anomaly. We have added this information in the revised version.

*Lines 282-284 page 10: This statement is not very convincing. Can we really say that the 9-14 km layer is a moist one when the observations show only a small peak around 10 km?*

We have corrected the text, and mention a peak at 10km and 15km.

*Line 320 page 12: I would really help to have a plot of the brightness temperature with the CALIOP track and the wind direction / geopotential to show the mean circulation pattern, same for the March case.*

The figure has been modified in the revised manuscript.

*Lines 505-508 page 18: The sentence, as is presented now, is not really giving an indication on the capability of the trajectory method in the convective origin study. Do you have some references indicating the quality of ECMWF 0.15x0.15 analysis in resolving vertical velocities for tropical cyclones? Also, I think this paragraph is better fitting in the method presentation of section 3.*

Recent improvements of the ECMWF IFS model have enhanced its forecasting skills of tropical cyclones (Magnusson et al., 2019). We have added this information in the revised manuscript, and the location of this paragraph has been modified.

*Lines 577 – 579 page 20: This sentence is confusing. Do you mean the difference averaged between 316 and 261 hPa is -20% for both days and for both CFH and MLS mean? That does not seem correct.*

We removed this discussion in the revised manuscript.

*Line 650 page 22: "The QBO easterlies can be observed at 70 hPa"…from here?*

This sentence has been removed.

**Technical corrections:**

*Line 15 page 1: It's worth to specify also in the abstract what CFH stands for.*

We removed the term CFH from the abstract and refer simply to balloon-borne measurements.

*Line 57 page 2: the upper 700m of what, the sea surface?*

700m of the ocean. The sentence has been corrected.

*Line 58 page 2 : .. that convection deeper than 15 km…*

Corrected

*Line 156 page 6: Do you mean: "..cross section. More details are given in the CALIOP Algorithm …"?*

Corrected

*Line 229 page 8: Latitude / longitude grid (5°x20° resolution)*

Corrected

*Line 230 page 8: at 215 hPa (figure 3) and 100 hPa (figure 4) for January 2016 (lower left panel) as March 2017 (lower right panel)*

Figure 3 and 4 are merged in a single figure in the revised manuscript.

*Line 269 Page 10: The red and purple lines…*

Corrected

*Lines 340-341 page 12: This is one example of a needed figure reference. Does it refer to figure 7?*

Yes. The figure is now properly referenced.

*Line 354 page 12: reference to bottom left panel of figure 7?*

Yes. The figure is now properly referenced.

*Line 650 page 22: ...the impact OF the 2016 strong ..*

Corrected

*Figure1: The green star on the grey background is not very easy to spot! Similarly for the dates label that are black and with a small font.*

The figure has been modified in the revised version

*Figure 5: Please, add a legend for the lines, it will ease the reading of the plot.*

A legend has been added

*Figure 10: The so called "brown dots" are difficult to distinguish from the red ones. The light pink is instead not very visible.*

The figure has been modified

*Figure 11:  The panels notation must be homogeneous. I would suggest indeed to reference the panels with letters, as done here, since it makes the reading more fluid. Same thing with the label "-1 day" "-2 days" that are missing on figure 10.*

The panels notation have been corrected

**Reference:**

Magnusson, L., J. Bidlot, M. Bonavita, A.R. Brown, P.A. Browne, G. De Chiara, M. Dahoui, S.T. Lang, T. McNally, K.S. Mogensen, F. Pappenberger, F. Prates, F. Rabier, D.S. Richardson, F. Vitart, and S. Malardel, 2019: ECMWF Activities for Improved Hurricane Forecasts. *Bull. Amer. Meteor. Soc.,* 100, 445-458, https://doi.org/10.1175/BAMS-D-18-0044.1

---

## Author Response (AR2)

**Reply to Anonymous Referee**

We thank the reviewer for her/his positive review. We have addressed the reviewer's comments in the revised manuscript. The modification are marked in red.

**Revised manuscript:**

[revised manuscript text omitted]